# Benchmarking an operational hydrological model for providing seasonal forecasts in Sweden

Marc Girons Lopez[1], Louise Crochemore[1,2], Ilias G. Pechlivanidis[1]

[1]Swedish Meteorological and Hydrological Institute, 601 76 Norrköping, Sweden
[2]INRAE, UR Riverly, 69100 Villeurbanne, France

*Correspondence to*: Marc Girons Lopez (marc.girons@smhi.se)

**Abstract.** Probabilistic seasonal forecasts are important for many water-intensive activities requiring long-term planning. Among the different techniques used for seasonal forecasting, the Ensemble Streamflow Prediction (ESP) approach has long been employed due to the singular dependence on past meteorological records. The Swedish Meteorological and Hydrological Institute is currently extending the use of long-range forecasts within its operational warning service, which requires a thorough analysis of the suitability and applicability of different methods with the national S-HYPE hydrological model. To this end, we aim to evaluate the skill of ESP forecasts over 39,493 catchments in Sweden, understand their spatiotemporal patterns, and explore the main hydrological processes driving forecast skill. We found that ESP forecasts are generally skilful for most of the country up to 3 months into the future but that large spatiotemporal variations exist. Forecasts are most skilful during the winter months in northern Sweden, except for the highly-regulated hydropower-producing rivers. The relationships between forecast skill and 15 different hydrological signatures show that forecasts are most skilful for slowly-reacting, baseflow-dominated catchments and least skilful for flashy catchments. Finally, we show that forecast skill patterns can be spatially clustered in 7 unique regions with similar hydrological behaviour. Overall, these results contribute to identify in which areas, seasons, and how long into the future ESP hydrological forecasts provide an added value, not only for the national forecasting and warning service but, most importantly, to guide decision-making in critical services such as hydropower management and risk reduction.

## 1 Introduction

Regardless of the geographical setting, human society depends on water resources to satisfy basic needs and allow for social growth and development. At the same time, however, the variability of the hydrological systems, leading to extreme events such as floods or droughts, puts pressure on the viability and sustainability of many water-intensive activities. In this setting, being able to predict the future evolution of the hydrologic system may improve societal resilience by anticipating potentially hazardous events and enabling the adoption of protective and/or adaptive measures (Girons Lopez et al., 2017; Pappenberger et al., 2015b). Even if most day-to-day decisions on water-related issues are based on short- and medium-range forecasts, some activities, such as water reservoir operation and optimisation or strategic planning, benefit from long-term forecasts (Foster et

al., 2018; Giuliani et al., 2020; Vigo et al., 2018). Despite their inherent uncertainties, such as hydro-meteorological model errors, future atmospheric states, or past hydro-meteorological water storages, long-term forecasts such as seasonal forecasts are a valuable tool for such applications, as they provide insights into the general trends of the hydrological system up to several months into the future, leading also to economic benefits (Bruno Soares et al., 2018; Giuliani et al., 2020).

Different techniques are available for generating seasonal forecasts, each with different strengths and weaknesses. These
techniques may be based on dynamic or statistical methods, or on a weighted combination of both. Among these, the Ensemble Streamflow Prediction (ESP) methodology – originally named Extended Streamflow Prediction (Day, 1985) – has long been widely adopted for seasonal forecasting (Wang et al., 2011; Wood and Lettenmaier, 2006). Following this methodology, ensemble streamflow forecasts are generated using historical meteorological data as forcing to a hydrological model. An advantage of this method compared to methods based directly on streamflow climatology, is that ESP forecasts are initialised
using the latest hydrological conditions (Crochemore et al., 2020), thus benefiting from the most recent hydrological knowledge when they are initialised, which is of particular interest for unprecedented hydrological conditions. This advantage can however also lead to forecast overconfidence as this method does not consider the impact of potential uncertainties in the initial hydrologic conditions, as noted by Wood and Schaake (2008). Additionally, the reliance of ESP forecasts on historical meteorological forcing makes it impossible for them to capture hydrological responses to unprecedented meteorological
events. Conversely, forecasts based on Numerical Weather Prediction (NWP) models are not constrained by the observational period as they are driven with an ensemble of dynamical meteorological forecasts, and are increasingly being used to overcome these limitations (Monhart et al., 2019). Nevertheless, ESP forecasts still offer the best study object to focus on the role of initial hydrologic conditions alone, which are best explained through catchment characteristics than by using NWP forcings.

ESP forecasts have been used by the scientific community to assess forecast skill sensitivity and uncertainties and to benchmark
seasonal forecast improvements (Arnal et al., 2018; Harrigan et al., 2018), as well as for operational flood forecasting in many different settings and scales (Candogan Yossef et al., 2017). Over the years, different techniques have been developed to improve the performance of forecasting systems, such as data assimilation for improving the initial conditions of forecasts (DeChant and Moradkhani, 2011), multi-model approaches (Muhammad et al., 2018), or pre- and post-processing techniques such as using artificial neural networks for reducing the effects of model errors (Jeong and Kim, 2005; Macian-Sorribes et al.,
2020), historical scenario selection and weighting (Crochemore et al., 2017; Trambauer et al., 2015), and calibration techniques (Wood and Schaake, 2008).

Evaluation efforts are typically carried out based on forecasts issued retrospectively (re-forecasts) over time periods long enough to ensure that the evaluation is statistically robust. For many operational applications it is important to understand the spatiotemporal patterns of seasonal streamflow predictability as well as the driving processes behind these patterns (Sutanto
et al., 2020). Indeed, previous studies have identified different sources of forecast skill depending on hydrological characteristics; for instance, Greuell et al., (2019), Shukla et al., (2013), and Wanders et al., (2019) identified initial conditions of soil moisture and snow (during spring) as the most important sources of skill over Europe, while Singla et al. (2012) found similar results for France. In a study over the United Kingdom, Harrigan et al. (2018) ascertained streamflow predictability

was higher for slow-responding catchments, as described by the baseflow index (BFI). Some studies have even gone one step further by investigating spatiotemporal patterns in streamflow predictability in an attempt to regionalize the forecast skill. For example, Pechlivanidis et al. (2020) showed that streamflow predictability is strongly dependent on the overall hydrological regime, with limited predictability in flashy basins (low river memory) and hence, it can be regionalised based on a priori knowledge of local hydro-climatic conditions.

The Swedish Meteorological and Hydrological Institute (SMHI) has long been operationally providing streamflow forecasts (catchment outflows) and hydrological warnings for Sweden (Figure A1). to relevant actors in hydrological risk management (municipalities, county boards, Swedish Civil Contingencies Agency), as well as to the general public. Additionally, both professional actors and the general public have access to the current hydrological situation and streamflow climatology through the open access *Vattenwebb* portal (https://vattenwebb.smhi.se/). On top of that, SMHI's consultancy services provide tailored forecasts to relevant actors. These forecasts are however not included in the public service and, as of today, are limited to individual river basins. Forecasts were initially produced with the HBV model (Bergström, 1976), but in recent years operational forecasting has shifted to the Swedish implementation of the HYPE model (S-HYPE, Lindström et al., 2010), which allows for an integrated, high-resolution description of the hydrological system across the country. Where available, in-situ observations of streamflow are assimilated, which has a beneficial impact on the hydrological predictions downstream. ESP seasonal forecasts are produced operationally but have not been widely used in real-world applications due to the lack of information on their skill and to the subsequent potential misinterpretation by external parties. Nevertheless, SMHI is now looking to extend the usage of long-term forecasts within its warning service, which requires a deeper understanding of forecast performance, its patterns, and controlling factors.

The aim of this study is to evaluate SMHI's operational ESP seasonal forecasts by benchmarking and attributing ESP forecast skill over Sweden with the operational S-HYPE model. To achieve these objectives, we: (a) evaluate the skill of ESP seasonal forecasts generated with the operational S-HYPE model over Sweden and understand the spatiotemporal pattern of skill, (b) detect potential links between streamflow forecast skill and hydrological characteristics, and (c) attribute streamflow predictability patterns across the country to hydrological behaviour of the river systems. The paper is structured as follows: section 2 presents the data used, hydrological model setup, and methodology for the forecast evaluation; section 3 presents the results, followed by the discussion in section 4; finally, section 5 states the conclusions.

## 2 Methods

### 2.1 Data

Daily precipitation and temperature data from the PTHBV database (Johansson, 2002) were used as forcing data to the S-HYPE model. This database contains gridded data based on a weighted interpolation from all available station observations for any given day with a resolution of 4x4 km, and it is available from 1961 onwards. The interpolation method used for generating PTHBV considers factors such as elevation and wind frequency and direction to make interpolated values for

precipitation and temperature more reliable. This dataset was processed using a weighted average method based on the area fraction of the PTHBV grid cells intersection with a given model catchment to force the semi-distributed S-HYPE model (Section 2.2). Additionally, daily stream discharge and water level data from 539 stations of SMHI's gauge network were used to correct the model outputs for improved forecast initialisation (see Figure 1). Data availability varies greatly among stations (Figure A2a). Nevertheless, most stations have observations for the entire study period (Figure A2b).

## 2.2 Hydrological modelling and forecasting

The ESP re-forecasts were produced using the S-HYPE model (Strömqvist et al., 2012), which is the operational implementation of the HYPE model for Sweden (Lindström et al., 2010). The HYPE model is a process-based hydrological model for water quantity and quality which operates on a daily time step and includes both hydrological (snowpack, groundwater, surface runoff, streamflow) and anthropogenic (reservoir operation, irrigation) factors. This model framework can be used in lumped, semi-distributed, and distributed modes. More specifically, the S-HYPE model is semi-distributed and, in its current version, consists of 39,493 catchments (with an average spatial resolution of 10 km$^2$) covering the whole of Sweden as well as parts of Norway and Finland in transboundary basins. The operational character of S-HYPE means that the model needs to perform adequately for a range of applications (e.g. early warning services, hydropower decision-making, water resources management) and it is therefore not parameterised towards a specific set of hydrograph characteristics. The model, which is continuously developed, has been calibrated and evaluated using a combination of numerical metrics such as the Nash-Sutcliffe Efficiency (NSE; Nash and Sutcliffe, 1970), the Kling-Gupta Efficiency (KGE; Gupta et al., 2009) and its components, in addition to multi-objective combinations of these metrics and expert judgement through visual evaluation and manual fine-tuning of model parameters. Focusing on the KGE metric, which is considered as a benchmark metric and provides information on the timing, volume and variability of streamflow, S-HYPE has a median efficiency of 0.79 for the period 1981 – 2016 at daily time step, ranging from -0.56 to 0.96 (Figure 1a). For reference, the KGE metric ranges between $-\infty$ and 1; the closer to 1 the KGE is, the more accurate the simulations are. This high KGE efficiency ensures that the aforementioned aspects of streamflow (timing, volume, and variability) are well represented by S-HYPE at most locations. Those stations showing poor KGE values generally correspond to highly-regulated or small catchments, where the timing and variability are more difficult to capture (Figure 1).

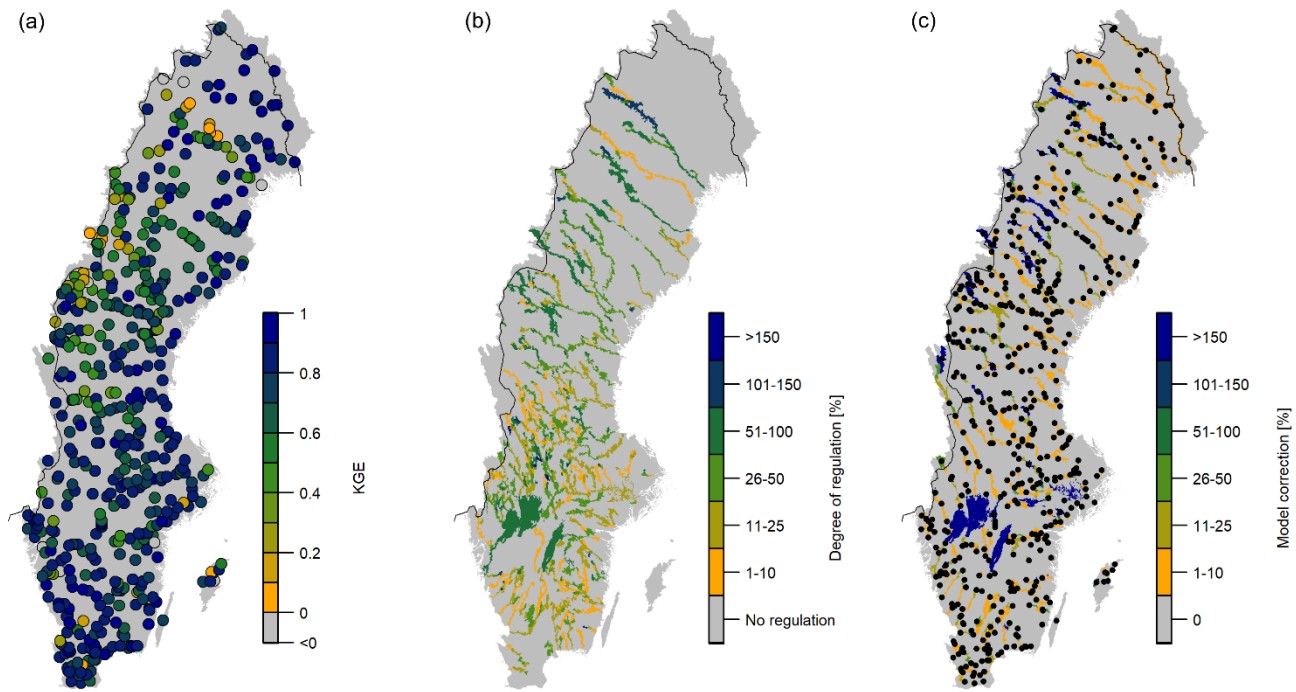

**Figure 1 Study domain: (a) S-HYPE model Kling-Gupta efficiency (KGE) for the period 1981 – 2016 for each of the 539 hydrological stations, (b) degree of regulation of each S-HYPE catchment, and (c) average model correction value for each S-HYPE catchment following the autoregressive correction method.**

A large percentage of water courses in Sweden are regulated, mainly for energy production purposes; see the degree of regulation (%) in Figure 1b; see also the definition in Pechlivanidis et al. (2020). This makes the simulation and prediction of water variables in the main water courses more challenging, as regulation patterns, which can largely deviate from the natural flow, need to be considered. In the operational S-HYPE model, general regulation regimes in the form of constant flow or seasonally varying sine wave shaped flow (or a combination of both) between predefined levels and, in some cases, specific dates are provided for a number of reservoirs. Nevertheless, since dam operation is continuously adapted (within certain bounds) to the changing meteorological and hydrological conditions, in addition to other factors such as optimising the economic benefit and ensuring safe operation, long-range forecasts based on hydrological models with only a limited description of such complex decisions on regulation patterns will most likely be conditioned by these simplifications.

We produced a series of hydrological re-forecasts up to a lead time of 190 days (approximately 6 months) at daily time step for all 39,493 locations across Sweden and transboundary basins using meteorological forcing data from 25 random years for the period 1961 – 2016 so as to mimic SMHI's operational setup (Figure 2a). When selecting the forcing data, a window of 3 years was left out around the analysis year (1 year before and 2 years after) to limit the impact of interannual streamflow memory and thus avoid conditioning the forecasts. We initialised the re-forecasts on the 1st, 8th, 15th, and 22nd of each month (approximately once a week) and aggregated the daily forecast data into weekly averages.

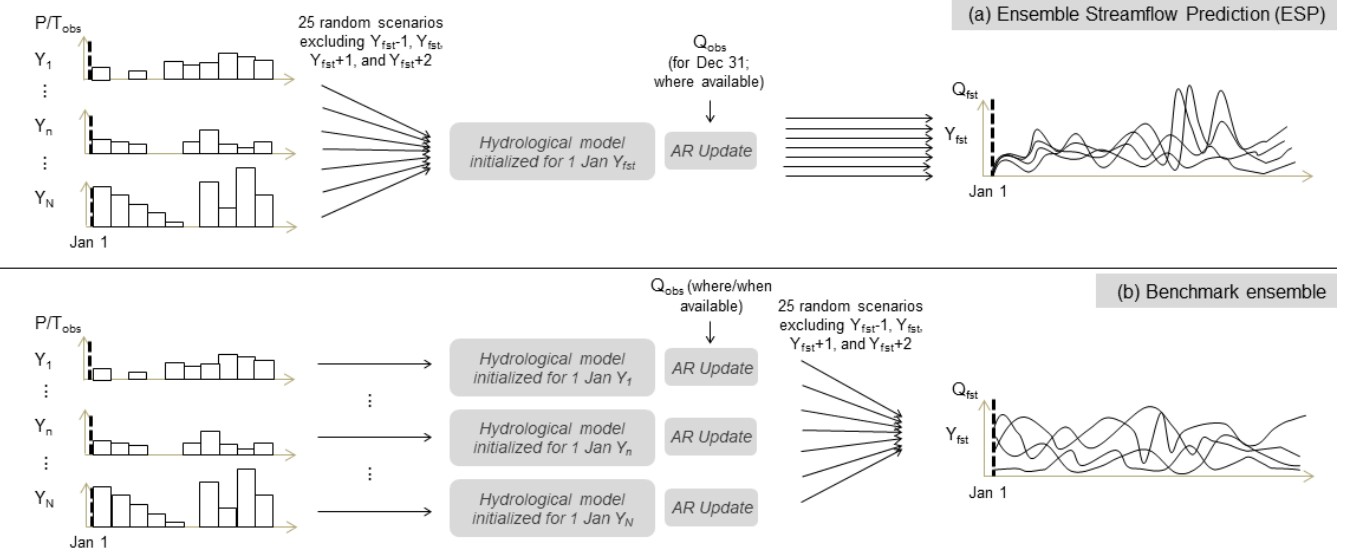


**Figure 2 Schematic of the forecast generation procedure in this study: (a) ESP re-forecasts and (b) benchmark forecasts. Adapted from (Crochemore et al., 2020).**

Following SMHI's operational setup, model outputs were corrected with stream discharge and water level observations, where and when available, to obtain the best possible initialisation conditions. When observations were no longer available, an

autoregressive (AR) correction method was used (Lindström and Carlsson, 2000; Pechlivanidis et al., 2014). To illustrate this procedure, let us consider the case of a catchment which has observations throughout the analysis period. For each forecast initialisation, the model outputs were corrected up to the day before forecast initialisation and model errors were stored. Then, as observations were theoretically no longer available, the model output correction started from the latest stored model error value and exponentially decreased with time based on a calibrated factor until the model outputs converged with the simulation

results. This correction only affects catchments with or downstream from observations and is especially relevant (at least for the first forecast lead times) for regulated water courses with low model performance where simulated streamflow can significantly deviate from actual values (Figure 1c).

In summary, the reforecast dataset has the following size: 39,493 catchments, 43,200 forecast initialisations (4 start dates per month x 12 months x 36-year reforecast period (1981 – 2016) x 25 members) using AR-correction where and when available,

and averaged weekly up to 190 days.

**2.3 Forecast evaluation**

We evaluated the skill of the ESP re-forecasts produced with the S-HYPE model over the period 1981 – 2016 using the Continuous Ranked Probability Skill Score (CRPSS, Appendix B) and a cross-validation strategy. Although studies involving

large-scale models often use model simulations as reference, as this minimizes the impact of model performance on forecast skill (Arnal et al., 2018; Crochemore et al., 2020), here we followed SMHI's operational setup and therefore used a reference based on a combination of observations (for catchments with or downstream from observation points) and model simulations (also known as perfect forecasts; elsewhere) to achieve the best possible initial conditions. We assessed the skill of the ESP re-forecasts so as to highlight the added value of the ESP forecasts with respect to an ensemble forecast based on streamflow

climatology, which users would have access to in the absence of SMHI's forecast service (Pappenberger et al., 2015a). To this purpose, we used an ensemble whose 25 members were resampled from the station-corrected streamflow climatology for the period 1981 – 2010 (excluding the forecast year) as a benchmark against which to derive the skill of the ESP re-forecasts (Figure 2b).

Even if hydrological models are typically run at a daily time scale, forecast results from hydroclimate prediction systems are

usually post-processed and aggregated over longer periods to provide information tailored to the user needs (Bohn et al., 2010). More specifically, a temporal aggregation of one month is typically used in seasonal forecasting services (Apel et al., 2018; Bennett et al., 2017). Nevertheless, different time periods may be of interest depending on the sectorial use (e.g. water resources management, civil protection mechanisms, warning services). Therefore, in addition to using a default temporal aggregation of one week to estimate the predictive skill of the national operational service, we were also interested in understanding how

aggregating streamflow forecasts over different time periods (i.e. 2 weeks, 4 weeks, 8 weeks, 12 weeks, and 24 weeks) would impact forecast skill at different lead times.

### 2.4 Forecast skill attribution

Thereafter, we investigated which hydrological characteristics are associated with skilful forecasts. More specifically, we selected a set of 15 hydrologic signatures (statistics describing the hydrological behaviour; see Table 1) to provide diagnostics

of the hydrological regime. Since no consensus exists an adequate set of hydrological signatures (McMillan et al., 2017), the set we used in this study draws on previous literature on hydrological classification (Euser et al., 2013; Viglione et al., 2013), process understanding (Kuentz et al., 2017; Pechlivanidis and Arheimer, 2015) and forecasting skill attribution (Pechlivanidis et al., 2020) and is based on the assumption that these signatures are not prone to large uncertainties and can thus provide information towards the identification of hydrologically similar river systems (Knoben et al., 2018; Westerberg et al., 2016).

We used the non-parametric Spearman rank test to assess the correlation between forecast skill and each of the hydrologic signatures.

**Table 1 Hydrologic signatures used for catchment functioning.**

| Signature | Abbreviation | Unit | Reference |
| --- | --- | --- | --- |
| Mean annual specific runoff | Qm | mm yr$^{-1}$ | Viglione et al. (2013) |

| | | | |
|---|---|---|---|
| Range of Pardé coefficients | DPar | - | Viglione et al. (2013) |
| Slope of streamflow duration curve | mFDC | % %$^{-1}$ | Viglione et al. (2013) |
| Normalised low streamflow | q95 | - | Viglione et al. (2013) |
| Normalised high streamflow | q05 | - | Viglione et al. (2013) |
| Coefficient of variation | CV | - | Donnelly et al. (2016) |
| Flashiness | Flash | - | Donnelly et al. (2016) |
| Normalised peak distribution | PD | - | Euser et al. (2013) |
| Rising limb density | RLD | - | Euser et al. (2013) |
| Declining limb density | DLD | - | Euser et al. (2013) |
| Normalised relatively low streamflow | q70 | - | Viglione et al. (2013) |
| Baseflow index | BFI | - | Kuentz et al. (2017) |
| Runoff coefficient | RC | - | Kuentz et al. (2017) |
| Streamflow elasticity | EQP | - | Sawicz et al. (2011) |
| High pulse count | HPC | - | Yadav et al. (2007) |

Then, we applied a *k-means* clustering approach (Jin and Han, 2011) within the 15-dimension space (hydrological signatures) to group the catchments into clusters based on similarities of basin functioning and further identify the dominant streamflow generating processes for specific regions. This is not the first regionalisation effort done for Swedish catchments. Indeed, four main hydro-climatic regions based on hydro-climatic patterns (Lindström and Alexandersson, 2004; Pechlivanidis et al., 2018) have typically been used for water management in Sweden. Nevertheless, this previous regionalisation was based on different
variables (e.g. marine basins) and is thus not suitable for the purposes of this study.

Finally, we analysed the hydrologic predictability for each of the clusters.

## 3 Results

### 3.1 Temporal and spatial distribution of forecast skill

The skill of the ESP re-forecasts varies with the lead time and the forecast initialisation date (Figure 3). As expected, the ESP
skill for 1-week forecast averages with respect to streamflow climatology, as measured by the CRPSS metric, is overall very high for medium-range horizons (i.e. 1 – 2 weeks ahead), with a median skill over Sweden starting at 0.7 (Figure 3a) and

thereafter decreasing with time (CRPSS ranges between 1 (best) and -∞). After approximately three months and until the furthest horizon (190 days), the ESP provides, on average, no added value with respect to streamflow climatology. Similar trends have been observed in other evaluations of forecasting systems over Sweden (Foster et al., 2018; Olsson et al., 2016). In particular, we note a rapid decrease in skill in the first forecast month (Crochemore et al., 2020; Harrigan et al., 2018). Consequently, under the common initialisation frequency of 1 month for many climate prediction systems (Batté and Déqué, 2016; Johnson et al., 2019), streamflow predictability is expected to remain low for periods beyond a 2-week forecast horizon. By increasing the frequency of forecast initialisation (e.g. from once a month to once a week), and hence frequently updating the initial hydrological states, it is possible to maintain a high streamflow forecast skill for extended forecast horizons (Figure 3b).

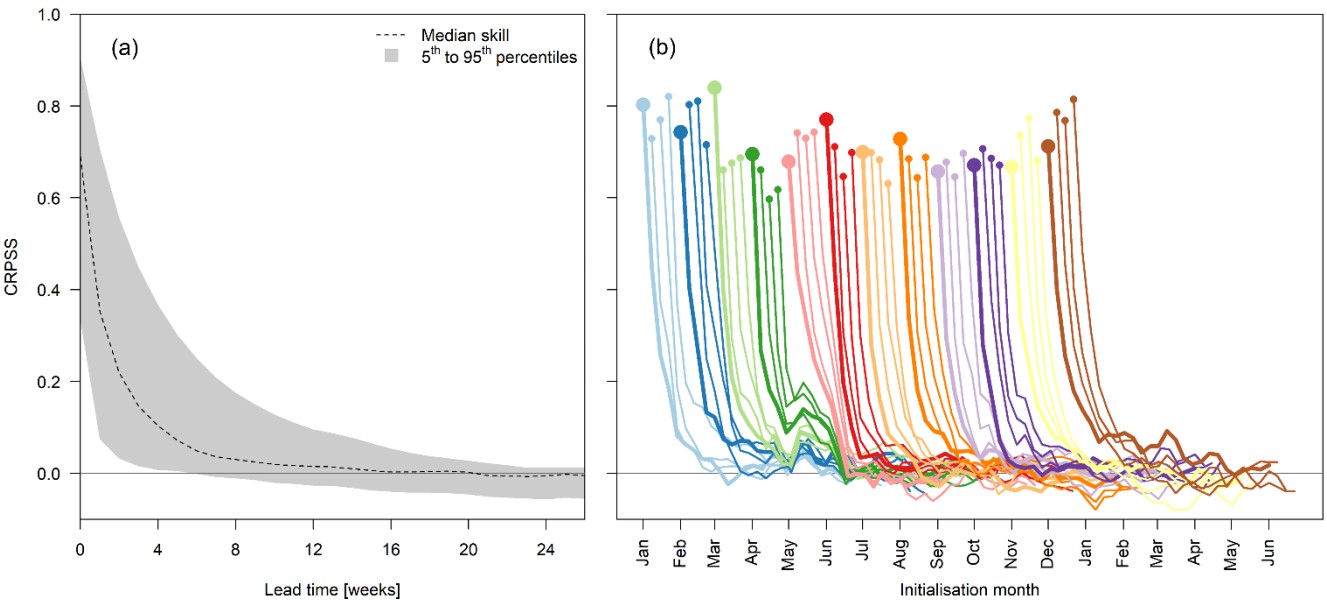

**Figure 3 Streamflow ESP forecast skill (in terms of CRPSS) for 1-week forecast averages as a function of lead time and up to 190 days: (a) median skill and 5[th] to 95[th] percentile range for the entire domain, and (b) temporal disaggregation of forecast skill per initialisation date. Initialisation dates within the same month (4 times) are represented with the same colour and the first initialisation of each month is marked with thicker markers and lines.**

Even if the forecast skill follows a similar decreasing pattern for all initialisation dates, both the maximum skill value as well as the deterioration rate differ. The highest skill (greater than 0.8) is observed for forecasts initialised between 8 December and 1 March, which roughly corresponds to the winter months. In the other seasons, the forecast skill starts at around 0.7, with the lowest skill value observed for initialisations in April (just under 0.6). Even if the forecast skill deteriorates quickly and reaches a predictability value close to the one of streamflow climatology (CRPSS close to 0) in long forecast horizons, forecasts initialised in March and April (and to some extent also in February) show a small secondary peak in the skill in May. This may be explained by the hydrological regime at a large part of Swedish catchments, in which streamflow generally starts to increase in April-May, despite the general lack of precipitation in winter and early March (see also Pechlivanidis et al., 2020).

The spatial distribution of forecast skill differs significantly across initialisation dates and forecast horizons (Figure 4). For instance, forecasts initialised in winter (e.g. December 1$^{st}$) maintain skill for inland forested areas of northern Sweden up to 3 months in the future. Forecasts initialised in spring (e.g. March 1$^{st}$) show skill up to the same forecast horizon, but most notably in the southern and eastern parts of the country. Finally, forecasts issued in summer and autumn are skilful up to 2 months except for some areas in the central-western parts of the country.

For the first forecast month, forecasts tend to have a comparatively poorer skill in the mountainous areas of north-western Sweden than in other parts of the country, except when they are initialised in the spring. Agricultural areas located around some of Sweden's largest lakes, such as Lake Mälaren and Lake Vänern (Figure A1), also have comparatively poor forecast skill. Interestingly, high predictability would have been expected in such lakes with slow hydrological response (long memory) (see Pechlivanidis et al., 2020). However, these great lakes are heavily regulated (see Figure 1), and the model correction seems to have impacted forecast skill. Streamflow forecasts in the large, highly-regulated, rivers of northern Sweden, such as River Umeälven and River Luleälven, also lack skill (Figure 1b; Figure A1). Again, the regulation patterns that significantly differ from the natural regime of watercourses, are not adequately captured by the ESP. In these cases, the broader ensemble of streamflow climatology is a better estimator of future streamflow. Conversely, streamflow forecasts show high skill in non-regulated rivers located in the same area and of similar size and hydrological regime, i.e. River Kalixälven and River Torneälven (Figure A1).

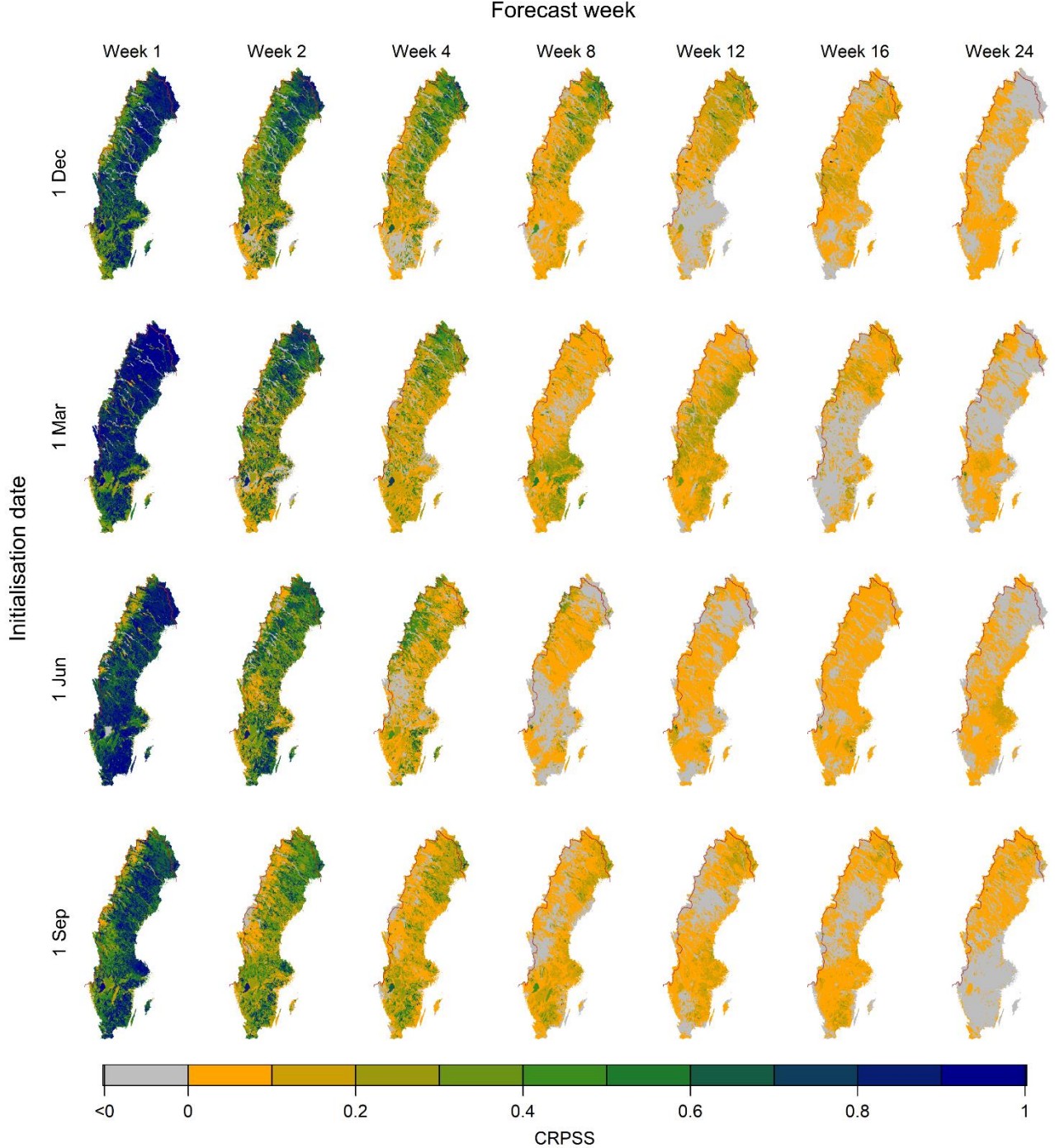


**Figure 4 Spatial distribution of streamflow forecast skill (in terms of CRPSS) for selected initialisation dates (rows) and forecast time horizons (columns).**

## 3.2 Forecast skill as a function of temporal aggregation

We next investigate the impact of using different forecast aggregation periods on the forecast skill for different lead times.
More specifically, in addition to the default aggregation period of 1 week, we consider the aggregation of streamflow forecasts over 2 weeks, 4 weeks, 8 weeks, 12 weeks, and 24 weeks. Since the focus here is not on the spatial patterns of skill, forecast skill is therefore averaged over the entire domain. Results show that, even if the average skill for the first forecast period decreases when aggregating over longer time periods, the forecasts remain skilful (CRPSS greater than 0) for aggregation periods up to 12 weeks (Figure 5). When aggregating over 24 weeks, the ESP method generally provides no added value with
respect to streamflow climatology; the predictability from ESP is very similar to the one from streamflow climatology. Even if, as expected, forecast skill decreases when forecasts are aggregated over long periods, a comparatively higher skill is maintained over longer time horizons than when forecasts are aggregated over short periods. In addition, forecasts initialised in February and March are skilful up to 16 weeks ahead when aggregating over long periods (e.g. 8 weeks), and the forecasts initialised in April and May show high skill values, even when aggregating over a 12-week period. This is probably due to the
high predictability of the spring flood season in May, also shown by the secondary skill peak observed for these initialisations in Figure 3b. Many catchments and rivers, especially in the northern half of Sweden, experience the peak of the spring flood during May. Using short aggregation periods, skill is more influenced by the exact start and end times of the spring flood event while long aggregations put more emphasis on a correct total flood volume. Since the total volume linked to the accumulated snowpack is easier to model than the timing of the event, which is conditioned by meteorological variables, long aggregations
tend to perform better. In the southern parts of the country, the spring flood is already over in May and low streamflow conditions start to dominate. Finally, for forecasts initialised in July and October, long aggregation periods (e.g. 12 weeks) tend to dilute the high forecast skill observed over the first weeks.

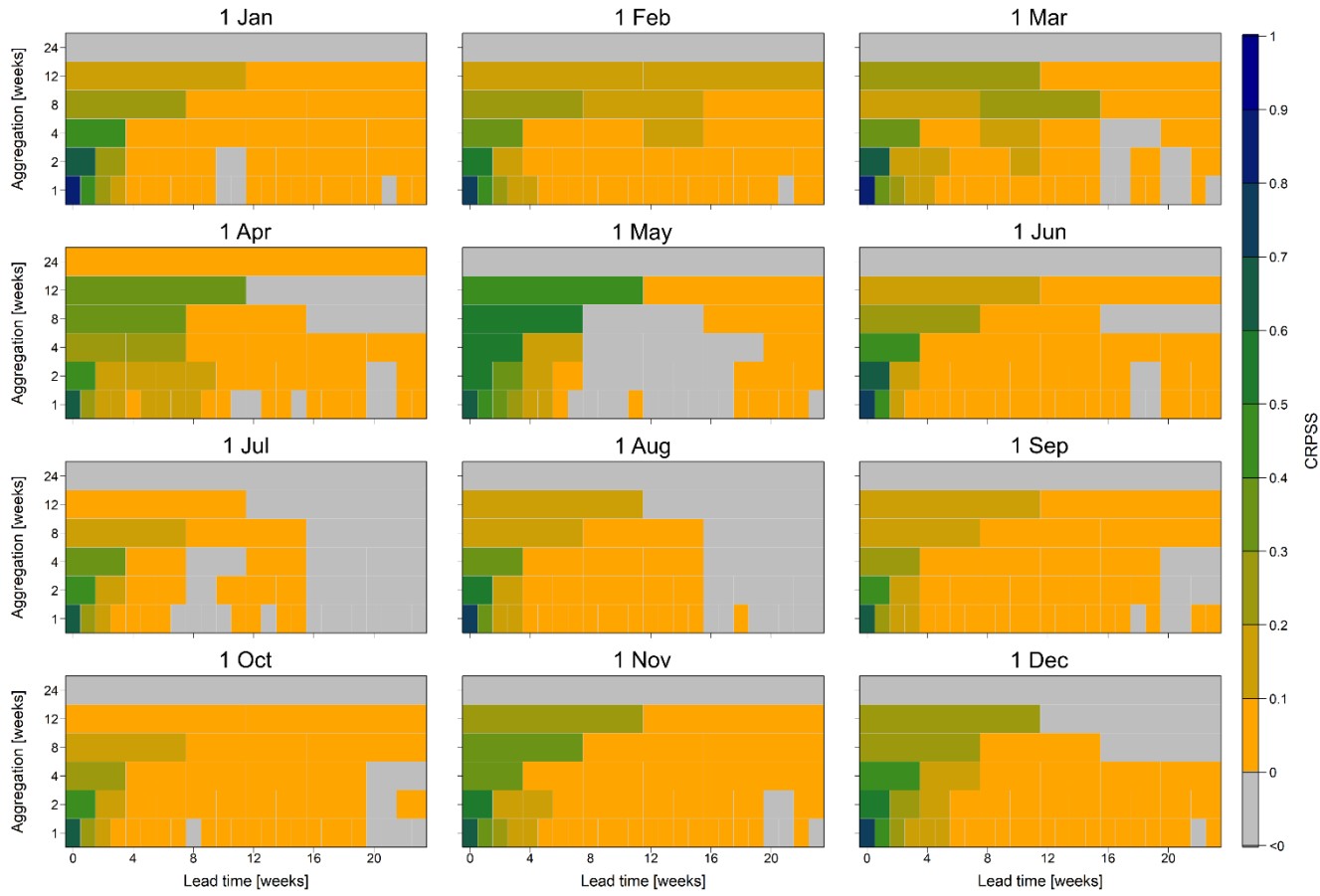

**Figure 5 Skill of streamflow forecasts as a function of lead time (weeks) initialised on the 1st of each month for selected forecast aggregation periods (i.e. 1 week, 2 weeks, 4 weeks, 8 weeks, 12 weeks, and 24 weeks). The skill is averaged over Sweden.**

### 3.3 Relating streamflow signatures and forecast skill

We next investigate potential correlations between forecast skill and the 15 streamflow signatures using the non-parametric Spearman rank test. In all cases the null hypothesis (i.e. no correlation exists between forecast skill and the streamflow signature) is rejected with a level of significance of 0.01 for lead week 0. Different patterns emerge when comparing forecast skill for each catchment with each of the 15 streamflow signatures (Figure 6). More specifically, forecast skill is strongly inversely correlated (defined here as the Spearman's rank correlation coefficient ($\rho$) being less than -0.50) with high pulse count (HPC), flashiness (Flash), rising limb density (RLD), declining limb density (DLD), and coefficient of variation (CV). Additionally, a strong direct correlation ($\rho > 0.50$) is found between skill and baseflow index (BFI), normalised low streamflow (q95), and normalised relatively low streamflow (q70) indicating that slowly reacting catchments with a significant baseflow component generally experience high predictability (Harrigan et al., 2018; Pechlivanidis et al., 2020). A similar analysis has been conducted for longer forecast horizons (e.g. lead week 8); however, since spatial patterns in forecast skill weaken and

blend in with the forecast horizon, the identified correlations do not have any explanatory power. Overall, the identified correlations highlight the existence of a generally high forecast skill in slowly reacting, baseflow-dominated catchments, while low forecast skill is predominant in flashy catchments. Although this analysis indicates the existence of dependencies between streamflow signatures and forecast skill, it can still be considered limited given that a hydrological system is generally characterized by a wider set of streamflow signatures than that considered here (Pechlivanidis and Arheimer, 2015; Sawicz et al., 2011).


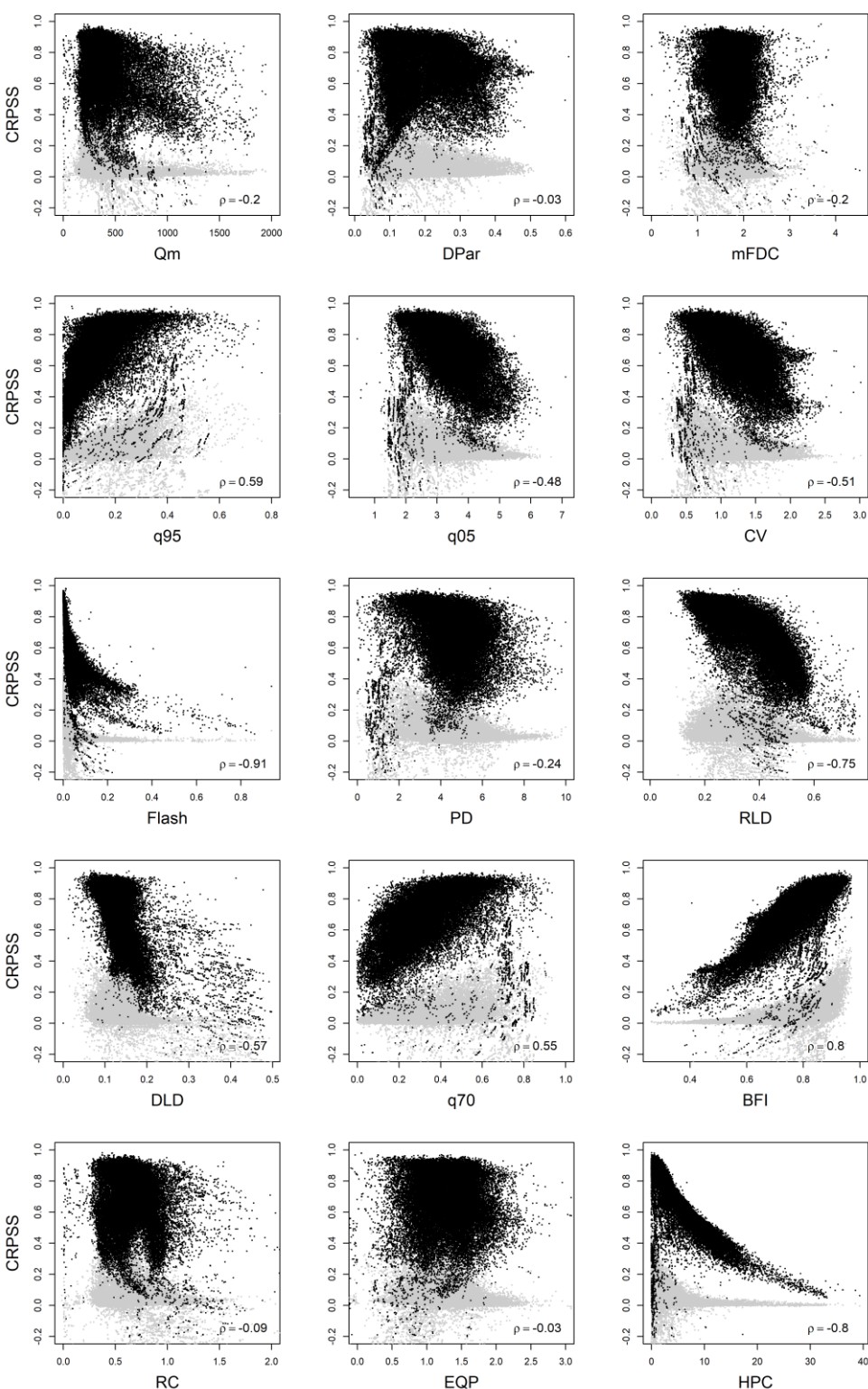

 **Figure 6 Forecast skill (in terms of CRPSS) for the lead week 0 (black dots) and lead week 8 (light grey dots) for each of the 39,493 catchments in Sweden as a function of each of the 15 hydrological signatures. The non-parametric Spearman's rank correlation coefficient for lead week 0 (ρ) is shown for each signature.**

### 3.4 Attributing streamflow forecast skill to hydrologic behaviour

Here, we investigate the potential attribution of streamflow predictability in the Swedish river systems to hydrological behaviour, given that such dependency has been highlighted in the previous analysis. Using the *k-means* clustering method,
an optimal number of seven distinct clusters (based on a silhouette analysis using a different number of clusters; de Amorim and Hennig, 2015) have been obtained representing different hydrological regimes (Figure 7). Table 2 provides additional information on the topographic, climatological and hydrological characteristics of each cluster while the spatial variability of each of the 15 streamflow signatures, as well as of the catchment elevation, is presented in Figure C1.

Catchments clustered in regions 1 and 5 are characterised by a high baseflow contribution (BFI), a slow response to
precipitation (Flash) and, therefore, a generally small intra-annual variability (DPar). In terms of topography, these regions consist of forested areas mainly located in southern Sweden. Catchments in cluster 2 are found in highland areas and boreal forest environments in northern Sweden, and are characterised by high seasonality (CV and DPar) due to the alternance between snow melting and accumulation. These catchments are also characterised by high runoff volumes (Qm) given that they are subject to high precipitation amounts and low evapotranspiration rates. Agricultural and coastal areas located mainly
in southern and central parts of the country are found in cluster 3. These catchments are characterised by a highly variable streamflow regime (HPC and RLD) and a quick response to precipitation (Flash), yet exhibit a relatively long hydrograph recession (DLD). Similarly, catchments grouped in cluster 6, which are located in low-land coastal and lake areas, experience flashy responses (Flash), as well as high streamflow (q05) and seasonal variations (CV and mFDC). Boreal forest catchments in the northern part of the country are grouped in cluster 4, and are characterised by a generally high runoff coefficient (RC)
and a slow response to precipitation events (Flash). Finally, catchments in cluster 7 are found along several large and highly-regulated rivers in northern Sweden. These catchments are characterised by a small variability (CV, DPar, and mFDC) but high streamflow volumes (Qm) and runoff coefficients (RC) explained by anthropogenic regulations.

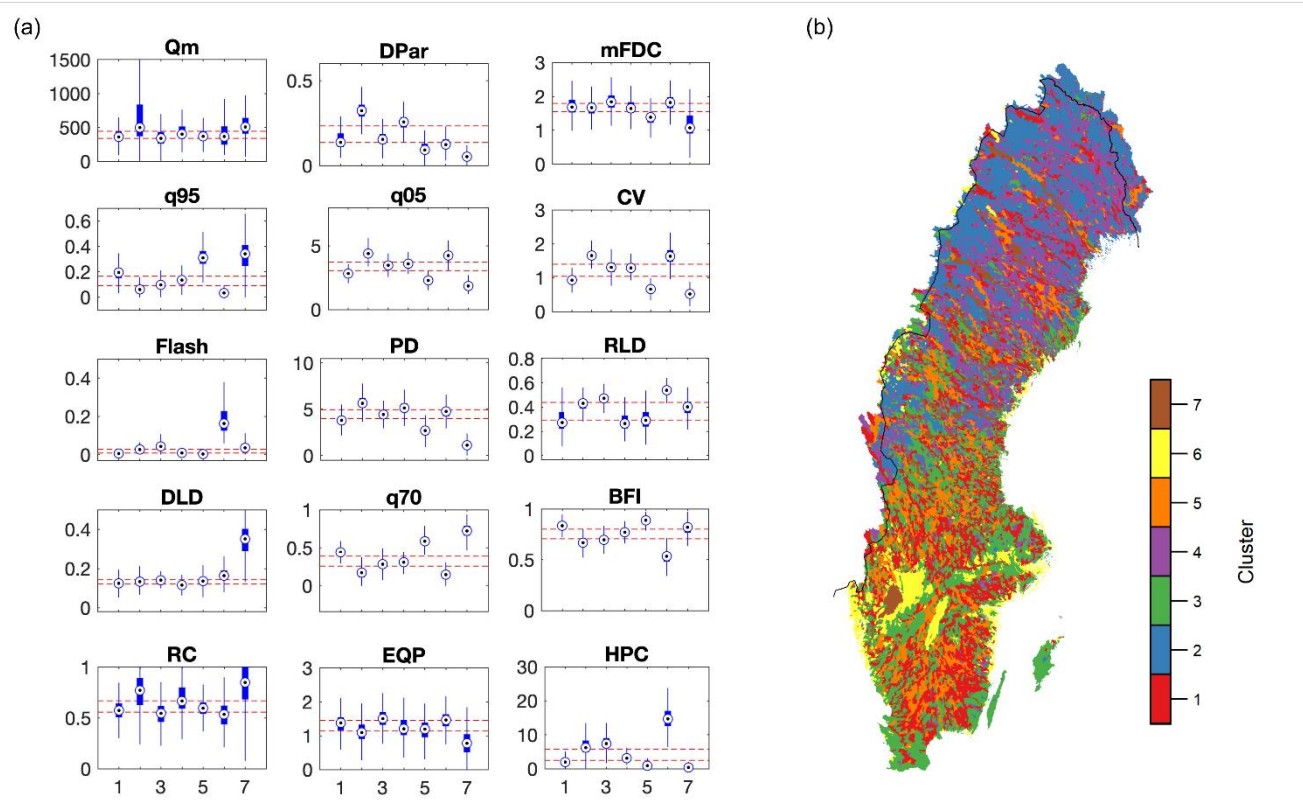

**Figure 7 Clustering analysis: (a) distribution of the 15 hydrological signatures in each clustered region. The red lines represent the 33$^{rd}$ and 66$^{th}$ percentiles for each signature; (b) geographical distribution of hydrologically similar regions over Sweden.**

**Table 2 Main characteristics of each of the hydrological clusters. The values provided for elevation, annual precipitation, and annual actual evapotranspiration correspond to the mean and interquartile range (25$^{th}$ – 75$^{th}$ percentiles).**

| Cluster region | Number of catchments | Elevation (m a.s.l.) | Annual precipitation (mm yr$^{-1}$) | Annual actual evapotranspiration (mm yr$^{-1}$) | Low streamflow signatures | High streamflow signatures |
|---|---|---|---|---|---|---|
| 1 | 8406 | 249 (109 – 344) | 650 (557 – 694) | 280 (232 – 334) | q05, CV, Flash, PD, RLD, HPC | q95, q70, BFI |
| 2 | 6705 | 478 (228 – 744) | 733 (581 – 901) | 175 (119 – 235) | q95, q70, BFI, EQP | Qm, DPar, q05, CV, PD, RC, HPC |
| 3 | 8250 | 165 (6 – 278) | 654 (552 – 704) | 294 (249 – 342) | BFI, RC | Flash, RLD, EQP, HPC |
| 4 | 8305 | 371 | 653 | 219 | Flash, RLD, DLD | DPar, PD |

| | | | | | |
|---|---|---|---|---|---|
| | | (214 – 490) | (560 – 678) | (180 – 263) | |
| 5 | 4329 | 266 (153 – 340) | 642 (556 – 693) | 270 (221 – 318) | DPar, mFDC, q05, CV, Flash, PD, RLD, HPC | q95, q70, BFI |
| 6 | 2405 | 31 (1 – 12) | 732 (544 – 824) | 306 (251 – 369) | DPar, q95, q70, BFI | mFDC, q05, CV, Flash, RLD, DLD, HPC |
| 7 | 1025 | 224 (99 – 321) | 619 (552 – 639) | 266 (233 – 294) | DPar, mFDC, q05, CV, PD, EQP, HPC | Qm, q95, DLD, q70, RC, |

The last step is to analyse the streamflow forecast skill in each hydrological cluster (Figure 8). Note that here we have aggregated the skill for all initialisations and hence we have not accessed the seasonal distribution of the forecast skill; however, we have focused on the detection of dependencies between skill and hydrologic regimes. Nevertheless, we note that the clusters with high (or poor) forecast skill in relation to the others are the same independently of the target month/week. According to Pechlivanidis et al. (2020), this is due to the intraannual variability of the streamflow response, which consistently varies between the catchments from the different clusters. Clusters 1, 4, and 5, which have high river memory due to baseflow domination, small intraannual variability, and generally low response to precipitation (see Table 2), have a higher median skill than the country-average. Among them, cluster 5 has the highest overall median skill for all time horizons but also, interestingly, the highest spread in forecast skill as a function of lead time. This may be attributed to the large variability in rising limb density (RLD) for the catchments in this cluster (see Figure 7a). The strong but negative correlation between forecast skill and RLD means that, despite the high baseflow contribution (BFI), some of the catchments in cluster 5 experience sharp increases in their hydrographs, which translates in low forecast skill. All of these clusters correspond mainly to forested catchments across the country. Cluster 3 and, most notably cluster 6, have a lower median skill than the country-average. These catchments are characterized by short river memory with flashy responses and are strongly driven by precipitation and strong seasonal variations. Similar results are observed for cluster 2. In this case, however, the median skill is closer to the country-average skill than for clusters 3 and 6. The response from catchments in cluster 2 is highly seasonal due to snow accumulation and melting processes, and hence not as rainfall-driven as for clusters 3 and 6. Finally, cluster 7, which contains the catchments along the large regulated rivers in northern Sweden, is the only set of catchments in which the median forecast skill reaches negative values, including also a large spread in the skill values ($5^{th}$ and $95^{th}$ percentiles). In these catchments, the ESP was expected to be outperformed by streamflow climatology since, as previously mentioned, the latter benefits from AR-correction throughout the forecast period and thus can better reproduce regulation patterns with low intraannual variability.

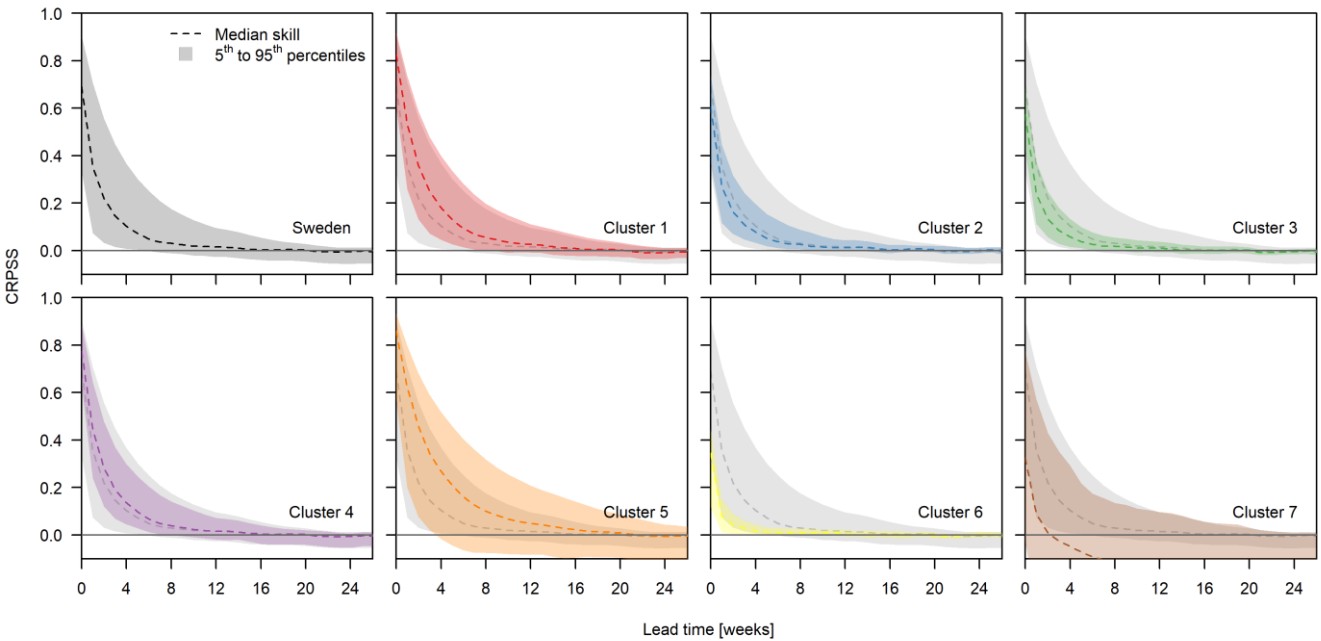

**Figure 8 Streamflow forecast skill (in terms of CRPSS) as a function of lead time for the entire country (top-left corner; also shown in Figure 3a) and each of the 7 clusters. The median skill and range for Sweden are provided in the background of clusters 1-7 to provide a reference for the values of each cluster.**

## 4 Discussion

### 4.1 Challenges and opportunities in an operational forecasting service

The results obtained in this study indicate that ESP seasonal forecasts produced with the operational S-HYPE hydrological model are skilful with respect to streamflow climatology on average up to 3 months ahead, despite the large temporal and spatial variabilities. This positive skill would make operational seasonal forecasts, in general, suitable to guide decision-making for applications requiring long-term planning, (e.g. water resources management, agriculture). Nevertheless, issues related to the modelling setup, the forecast methodology, the hydro-climatic characteristics of the Swedish river systems (e.g. high degree of regulation in many water courses), among others, can impact the reliability of such a forecasting service.

The ESP forecasting approach is limited by its use of historical meteorological forcing data to generate the streamflow forecasts, making it unable to capture unprecedented meteorological events. Consequently, extreme events that lay outside the observed range will inevitably be misrepresented, limiting the service's predictability of extreme conditions which can be important to some decision makers. This issue may be addressed by using NWP models to predict the future climate (Monhart et al., 2019). However, although NWPs are not constrained by the observational period, they are limited by the chaotic nature of the weather system (aleatory uncertainty), which makes small errors in the initial conditions grow significant with time. In addition, NWP-based forecasts require post-processing (i.e. downscaling and bias-adjustment) to be suitable to use in impact

studies. Finally, their added value for streamflow forecasting in comparison to ESP is shown to be limited in Sweden, with the possible exception of southern Sweden (Arnal et al., 2018).

As expected, ESP forecast skill decreases rapidly with time particularly in fast responding river systems. Results showed that monthly initialization, which is the most common initialization frequency of climate prediction models, is critical to set high skill values; however, such frequency in the initialization cannot account for skill deterioration within the month. In this setting,
increasing the initialisation frequency to, for instance, once a week would allow to maintain a high skill up to monthly time horizons. Nevertheless, considering that climate prediction models are not developed to represent the exact daily dynamics of the natural systems and that forecasts are therefore aggregated into long time periods, more frequent (e.g. daily) forecast initialisations are not expected to provide an added value to the forecast service in terms of useful information for decision-making at seasonal horizons since long-term decisions are in any case not taken daily. Moreover, forecast information from
such frequent initialisations can easily be misinterpreted by decision-makers (Schepen et al., 2016).

Regarding the aggregation of forecast outputs, most studies have focused on a 1-month aggregation period as it is reported to provide an *"appropriate forecast at the seasonal scale and a proxy of the underlying distribution"* (Emerton et al., 2018; Meißner et al., 2017; Yossef et al., 2013). Nevertheless, since the way a seasonal forecasting service is used in decision-making depends on the sector, user, and service properties, there may be value in considering aggregation periods different from the
standard monthly aggregation (or even adaptive aggregation periods) for providing guidance on the usability of the forecasts for decision-making. For instance, for the energy sector, Swedish hydropower companies tend to be interested in a fixed 3-month aggregation over the period May-July. Alternatively, crop water needs can be assessed over the entire summer season to get estimates of required water volumes for irrigation. The produced matrix for different aggregations, initialisations, and lead times (Figure 5) allows communication of skill to various users depending on their needs. Our findings suggest that
aggregations over periods longer than the default 1-month do not necessarily mean a loss in skill. On the contrary, here we observed that, in Sweden, long aggregations of streamflow forecasts covering the spring flood season tend to gain in skill. Overall, however, from time horizons of, on average, 4 months into the future, forecasts have very low or no skill regardless of the aggregation period of choice.

Another important factor driving hydrological predictability at the seasonal scale is the adequate knowledge of the initial
hydrological conditions (Shukla et al., 2013). In many cases, ESP forecasts are initialized based on the latest available model state (modelled reality), which may significantly deviate from the actual hydrological state (observed reality). Incorporating the latest available observations into forecast initialisation can thus be especially important to bridge the gap between modelled and observed reality. Here, the model outputs were corrected whenever observations were available (and using AR-correction thereafter), with the objective to generate forecasts which are as close as possible to observed reality (see Section 2.2). This
method is straightforward and easy to implement, and takes advantage of streamflow memory to not only correct the initial forecast state but also the following forecast horizons when observations are no longer available. More advanced data assimilation methods could be considered in further developments of the presented operational forecast system, such as Kalman Filters (Sun et al., 2016), allowing not only for a correction of model outputs, but also an adjustment of model states

and thus of process representation (Musuuza et al., 2020). Additionally, observations other than streamflow, such as soil moisture or snow water equivalent, could also be assimilated into the model (Huang et al., 2017; Musuuza et al., 2020). Regarding snow water equivalent, snow is a key component of the hydrological cycle in many Swedish catchments and therefore, the impact of snow accumulation and melting on ESP forecast skill would deserve further investigation. Snow processes play an important role in river memory together with other processes such as groundwater/baseflow contribution, or hydrograph dampening from lakes. Contrary to the other two, however, snow processes tend to define the catchment dynamics only seasonally (e.g. precipitation in the form of snow in early December may be accumulated and further released as meltwater during the spring flood period), and hence the role of snow on ESP forecast skill is expected to have a seasonal pattern too.

Another approach to obtaining updated knowledge on the initial hydrological conditions is through frequent forecast initialisation. Our findings suggest that using weekly forecast initialisation instead of the more common monthly initialisation may significantly improve the streamflow ESP forecast skill, which is expected to add value to decision-making in different contexts. This may be of particular importance for periods in which decisions are subject to hydrological responses that alter in a short time window. For instance, in Sweden it is important to be able to predict the onset of the spring flood due to a combination of snow melting and precipitation, and adjust the reservoir regulation accordingly to optimise the power production for the coming months.

Different components of the S-HYPE modelling and forecasting chains, such as the model setup, forcing data, model structure and model parameters, lead to uncertainties in the forecasts results. Moreover, the setup used in this study, which uses a combination of observations and perfect forecasts as reference, makes the assessment of these uncertainties particularly complex. The contribution of model error to the total uncertainties in the results obtained here is removed from those catchments in which forecasts are evaluated against perfect forecasts. This non-represented contribution of model errors can nonetheless be considered minimal due to the high KGE performances of S-HYPE (see Figure 1a), which ensure a fair representation of temporal dynamics in non-regulated Swedish rivers. However, these errors may become significant for catchments with – or downstream of – observations, especially due to the interplay between correction of model outputs with observations and streamflow regulation. While model outputs are corrected with all available observations, not all watercourses with observations are regulated, and even those that are regulated do not necessarily have observations downstream from dams or other river regulation structures. The correction of model outputs with observations and, when these are no longer available (e.g. beyond forecast initialisation), with an exponentially decreasing factor based on the last known model error (i.e. AR-correction) may effectively reduce corresponding uncertainties, especially in the first time steps of the forecast. The downstream distance of a given catchment with respect to an observation is also relevant in this case, as model correction will only affect a fraction of the streamflow forecast at that location. The largest uncertainties, though, can be expected for heavily-regulated catchments with or downstream of observations. In these locations, complex river regulation routines, which depend on factors external to hydrological models, make it almost impossible for these models to adequately reproduce streamflow dynamics. Consequently, even if the correction of model outputs with observations may minimise uncertainties at forecast initialisation, these errors will rapidly spread due to the inability of the model to reproduce the modified hydrological regime.

## 4.2 Impact of regulation on forecasting skill

One of the main applications of long-term forecasting in Sweden is for planning reservoir operation during the spring flood season (May-July) (Foster et al., 2018). However, forecast skill is low for the main hydropower-producing heavily-regulated rivers in the northern parts of the country, where the highest spring flood peak volumes occur. Nevertheless, high ESP forecast skill, and subsequently valuable forecast information, can still be expected in the upstream reaches of these rivers that are not affected by other upstream regulation. Following this assumption, in these locations, even if ESP forecasts have no value for predicting reservoir outflows with respect to using the ensemble of streamflow climatology, they may adequately predict the water inflows from the headwaters to the reservoirs. In order to further understand the impact of streamflow regulation on the results, we evaluated the ESP forecasts using model simulations (without AR-correction) as reference. Forecast skill was in this case very high for the highly regulated rivers where low forecast skill was obtained in the main analysis. This exercise shows that the regulation routines in some river stations in the S-HYPE model still need improvement in order to correctly represent the management rules dominating regulated streamflow patterns. This issue is not as obvious in less heavily regulated rivers elsewhere in the country, where ESP forecasts are generally skilful.

With the exception of River Luleälven and other comparatively smaller rivers in the Swedish mountains, the S-HYPE model performance is generally high for most locations, including the large rivers in the northern parts of the country. Similarly, ESP seasonal forecasts are skilful for non-regulated rivers in that area that also benefit from long-term planning. More specifically, River Torneälven and, to a lesser extent, River Kalixälven are susceptible to severe ice break-up events in connection to the spring melt season and subsequent spring flooding (Zachrisson, 1989). An important factor in predicting the timing of the ice break-up is the onset of spring flood due to snowmelt. Skilful ESP seasonal forecasts for these rivers should allow for early planning and allocation of resources that could greatly contribute to mitigate potentially severe ice break-ups.

## 4.3 Regionalisation of skill in other domains

Besides streamflow regulation patterns, certain characteristics of the hydrological regime have a high impact on hydrological predictability. Here we have shown that forecast skill is high in baseflow-dominated catchments where past hydrologic conditions drive the catchment response, while it is low in flashy catchments where rainfall drives the streamflow dynamics and hence accurate rainfall forecasts are crucial. This corresponds well with findings from similar studies over different geographical domains (Harrigan et al., 2018; Pechlivanidis et al., 2020). However, contrary to the findings by Harrigan et al. (2018), who identified a specific streamflow signature (i.e. the baseflow index) as the main driver of hydrological predictability, we have found that, for Sweden, it is instead the result of the overall hydrological behaviour, even if some specific streamflow signatures may have a greater impact than others. The exact magnitude of the impact of the different signatures is however difficult to quantify since, even if no consensus exists on an representative set of signatures (McMillan et al., 2017), one can argue the hydrological system is generally characterised by a wider set of streamflow signatures than that considered here. In this context, we hypothesised that the selected signatures are robust enough to describe the hydrological

regimes and further guide the analysis towards the identification of hydrologic similarities. Additionally, the seven clusters not only differ in terms of hydrological response, but also in terms of climatological patterns and physiographic characteristics. The results obtained here may contribute to guiding in which areas, seasons, and how long into the future ESP hydrological forecasts provide an added value not only for SMHI's forecasting and warning service, but most importantly for guiding decision-making in critical services such as hydropower management and risk reduction. Here, we note that, even if the hydro-climatic gradient of Sweden does not fully represent the equivalent gradients over the continent or the globe, our results are however transferable to other locations with similar climatological and hydrological conditions as it has also been highlighted in Pechlivanidis et al. (2020).

## 5 Conclusions

Herein, we analysed the skill of ESP re-forecasts using the operational S-HYPE hydrological model over Sweden in an effort to evaluate the suitability of this methodology for producing skilful forecasts at the seasonal scale within SMHI's hydrological forecasting and warning service as well as for other activities requiring long-term planning. In addition, we aimed at understanding the underlying patterns and drivers behind skilful forecasts and attributed the seasonal predictability to hydrological characteristics. Approximately 39,400 catchments, lying along Sweden's strong hydroclimatic gradient, were investigated. The main conclusions of this study are:

- The skill of the ESP forecasts varies both geographically and seasonally, and depends on the initialization month and aggregation period. Moreover, the skill decreases rapidly with time particularly in fast responding river systems; however, the ESP forecasts are generally skilful up to 3 months into the future. Forecasts are most skilful during the winter months for the northern parts of the country, except for the highly-regulated hydropower-producing rivers.

- Initialization frequency is a key driver affecting streamflow forecasting skill. Monthly initialisations are critical to preserve high forecast skill values without, however, addressing the skill deterioration over the first forecast month. Increasing the initialisation frequency to once a week allows maintaining the high skill up to monthly time horizons.

- The river systems in Sweden can be categorised into 7 clusters based on similarities in streamflow signatures. This results in an improved understanding of the dominating hydrological processes, which are shown to vary spatially and seasonally. Particularly, dominant streamflow generation processes over the mountainous regions, including baseflow and snow accumulation/melting, dampening from lakes, and reservoir alterations could explain the hydrological clustering across the country.

- A link between forecast skill and streamflow signatures has been detected. Over the 15 streamflow signatures investigated here, baseflow index, flashiness, rising limb density, coefficient of variation and high pulse count show strong correlations with forecast skill. Streamflow forecasts are most skilful for slowly-reacting catchments due to snow-related processes and/or dampening from lakes and baseflow-dominated catchments (river systems with long

memory). Conversely, forecasts are least skilful for catchments with a flashy response to rainfall (river systems with short memory).

**Appendix A: Study domain and data availability**

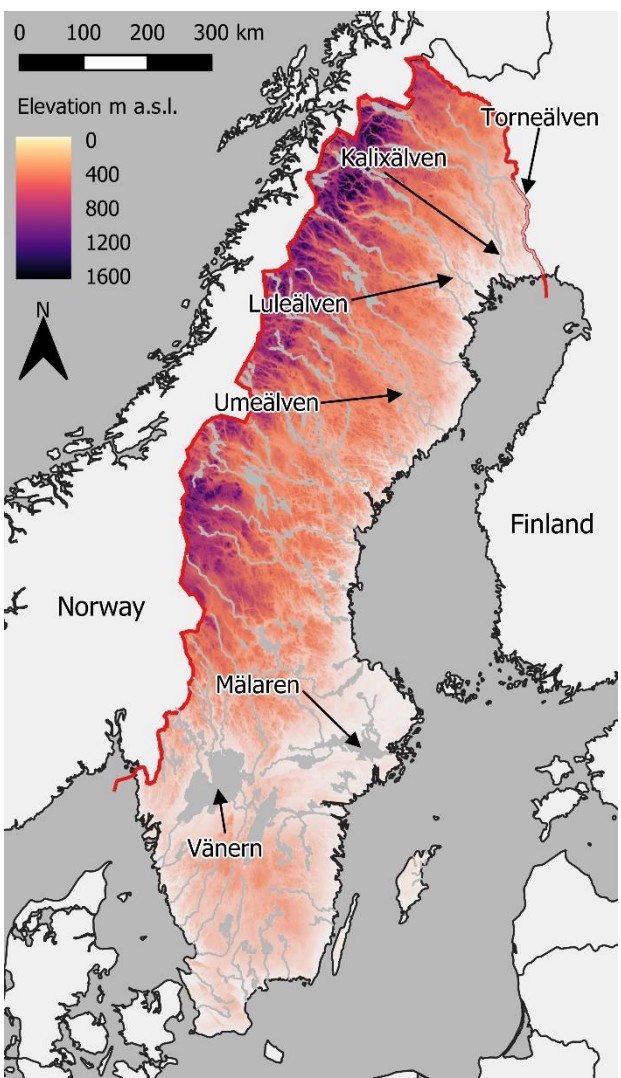

490 **Figure A1 Map of Sweden showing its topography and main hydrographic network. The rivers and lakes referred to in the main text are indicated here for convenience.**

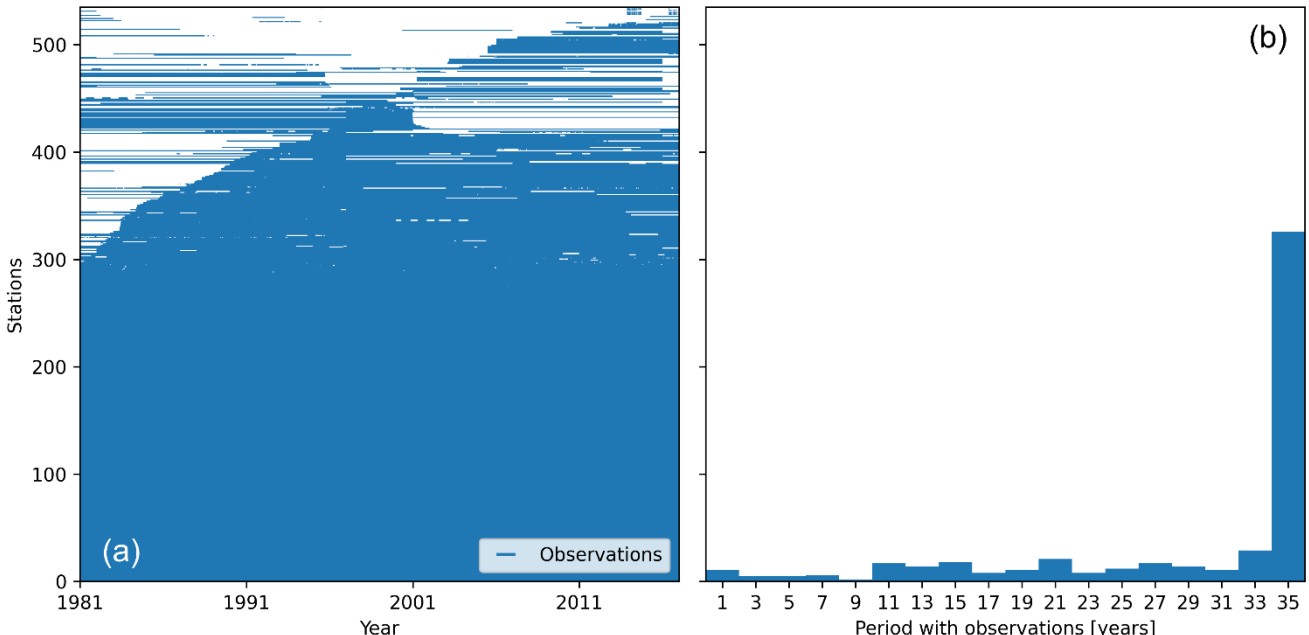

**Figure A2 Streamflow observations used in this study: (a) Temporal availability of observations for each of the 539 stations; sorted from longer (bottom) to shorter availability (top); (b) Histogram showing the total number of years of observations for all the stations.**

## Appendix B: Continuous Ranked Probability Skill Score

The Continuous Ranked Probability Score (CRPS; Hersbach, 2000) is a common measure of ensemble forecast performance. It is formulated as the integral squared distance between the forecast ensemble and the observation step function. The CRPS is then averaged over all forecasts of the evaluation period. Its dimension is that of the forecast variable being assessed, here $m^3\ s^{-1}$, and its value is equivalent to the mean absolute error when applied to deterministic forecasts.

The Continuous Ranked Probability Skill Score (CRPSS) is then assessed by comparing the CRPS value of the investigated forecast system (here, ESP) to that of a selected benchmark (here, an ensemble of streamflow climatology selected from the period 1981 – 2010). Given $CRPS_{sys}$, the CRPS of the forecasting system, $CRPS_{bench}$ the CRPS of the benchmark and $CRPS_{pft}$ the optimal CRPS value (0), the CRPSS is formulated according to Eq. A1.

$$CRPSS = \frac{CRPS_{bench} - CRPS_{sys}}{CRPS_{bench} - CRPS_{pft}} \tag{A1}$$

This metric is non-dimensional and takes values between 1 (optimum) and $-\infty$. Positive (negative) skill scores indicate that the forecast system performs better (worse) than the benchmark in terms of CRPS. Skill scores close to 0 indicate that the evaluated forecast system has equivalent performance to that of the benchmark.

**Appendix C: Spatial variability of hydrological signatures**

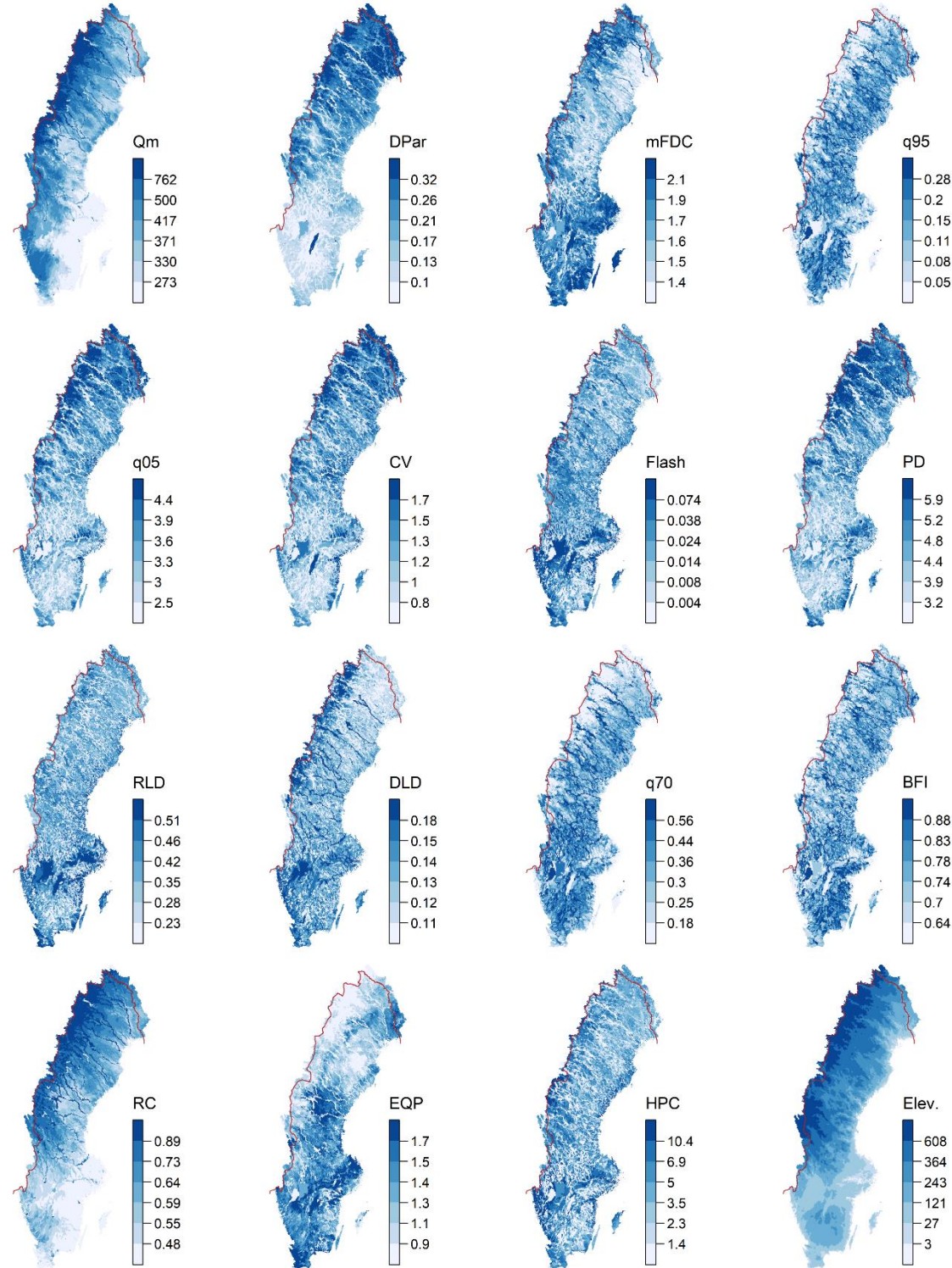

510

**Figure C1 Spatial variability of the 15 modelled hydrological signatures including the catchment mean elevation. The colour intervals are based on the quantiles (15% intervals) of each signature (and elevation) distribution. A clarification of the abbreviations used here can be found in Table 1 in the main text.**

## Data availability

The HYPE model code is available from the *HYPEweb* portal (https://hypeweb.smhi.se/model-water/). The meteorological data used for driving the ESP re-forecasts (PTHBV) is available from the *LuftWebb* portal (https://luftweb.smhi.se/), and the hydrological data used for model correction is available from the *Vattenwebb* portal (https://vattenwebb.smhi.se/).

## Author contribution

M.G.L. contributed with the study design, model runs, result analysis and figures, interpretation of the results, and writing the manuscript; L.C. contributed with the study design, code development for post-processing of results, the interpretation of results, and writing the manuscript; I.G.P. was responsible for the project management and funding acquisition, and contributed with the basic idea, the study design, clustering analysis and figures, interpretation of the results, and writing the manuscript.

## Competing interests

The authors declare that they have no conflict of interest.

## Financial support

This work was funded by the project "Long-term forecasts of wind and hydropower supply in a fluctuating climate – Importance for production planning and investments in energy storage and power transmission" granted by the Swedish Energy Agency under grant agreement No. 46412-1. Funding was also received from the EU Horizon 2020 project S2S4E (Subseasonal to seasonal forecasting for the energy sector) under Grant Agreement 776787. This study was also partially funded by the EU Horizon 2020 project PrimeWater (Delivering advanced predictive tools from medium to seasonal range for water dependent industries exploiting the cross-cutting potential of EO and hydroecological modeling) under the Grant Agreement 870497.

## Acknowledgements

The authors would like to express their sincere gratitude to Jim Freer, Louise Arnal, Shaun Harrigan and an anonymous reviewer for their constructive comments.

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
