# Peer review of "Benchmarking an operational hydrological model for providing seasonal forecasts in Sweden"

_Hydrology and Earth System Sciences, 2020_

## Referee Comment (RC3)

Girons Lopez et al. evaluate the forecast skill of the Ensemble Streamflow Prediction (ESP) method using the SMHI operational configuration of the S-HYPE hydrological model for providing seasonal streamflow forecasts across Sweden. They generate a set of ESP reforecasts at 25 ensemble members each for 39,493 catchments, initialised 4 times per month over the 36-year period from 1981-2016. The hydrological model, and reforecasts, are run at a daily time-step and are primarily aggregated to weekly averaged streamflow, out to a 6-month forecast horizon; a number of different temporal averages are also tested, from weekly to 24 weeks. For the 539 catchments with streamflow and water lever observations, an additional simple autoregressive algorithm was applied to correct raw modelled streamflow output prior to generation of the reforecasts. The probabilistic skill of ESP reforecasts was benchmarked using the Continuous Ranked Probability Skill Score (CRPSS) against a probabilistic (25 ensemble member) streamflow climatology (called "historical streamflow" in the paper) with modelled streamflow simulations as proxy observations (called "modelled reality" in the paper), or where available (i.e. 539 stations) in situ streamflow observations (called "observed reality" in the paper), as reference. Results show that ESP is skilful up to 3 months ahead for the most of Sweden. The strength of skill varies widely across the country in space and time and has shown to be linked to a number of key hydrological signatures (15 explored in total). Similar to previous work, ESP skill was highest in slowly responding baseflow-dominated (or high BFI) catchments and least skilful for flashy catchments. Seven unique clusters of similar hydrological behaviour were identified using k-means clustering and ESP skill summarised for catchments within each cluster.

I found this paper very interesting with a comprehensive ESP reforecast experimental design (i.e. long reforecast period, many forecast start dates, large sample of catchments, and cross validation used). It has the clear purpose of providing the scientific foundation for when and where the ESP forecast method is, and importantly is not, appropriate to use in operational seasonal forecasting across Sweden. The paper goes on to explore the potential sources of ESP skill based on correlation with hydrological signatures, while it's arguably a stretch to call correlation a formal attribution, the analysis nonetheless reveals interesting drivers and patterns of skill across the country, including the poor skill from catchments with high human disturbance (e.g. reservoirs). The paper is well structured and written with very good Figures. The analysis presented in Figure 7 is particularly insightful, and an innovative way of presenting forecast skill by clusters of similarly responding catchments. I offer below suggestions on areas where the paper could be expanded and highlight where clarifications are necessary, but these are all minor.

Therefore, I strongly recommend Girons Lopez et al. to be published in HESS. It adds to the growing literature benchmarking the skill of the ESP method with a clear application within operational seasonal forecasting at the national scale in Sweden.

**Main comments**

**1.)** It would be useful to have these parts of the methods expanded/clarified:

 a. **Pg3 L89-90:** While there is a link to the general website to download the streamflow observations in the "Data availability" section, there is little detail for the reader on if all 539 stations are available in near-real time, which would be necessary to understand the transferability of forecast skill results to operational forecasts in future. Also, are all stations available for the full 1981 to 2016 period for calculation of KGE in Fig. 1 and for calculating the historical streamflow benchmark forecast? Were most of all these stations used for calibrating the configuration of S-HYPE used in the study?

b. **Pg 5 L116-123:** I find the AR correction interesting, but there is very little detail on how it was applied within the current experimental design, and perhaps even if it was implemented in such a way that is as consistent as feasibly possible to the configuration that is/will be implemented operationally?

c. **Pg 5 Sect. 2.3:** The exact reforecast size is mixed between Sect. 2.2 and Sect. 2.3 and the reader has to try piece it together, it would be good if summarised. My understanding is that the reforecast dataset used has the following size: 39,493 catchments; 1728 start dates (4 start dates per month x 12 months x 36-year reforecast period (1981-2016)); weekly averaged streamflow out to 6 month forecast horizon at 25 ensemble members each?

d. **Pg 5 L130-134:** I think it could be confusing to refer to the probabilistic streamflow climatology benchmark forecast as "historical streamflow" because historical streamflow could more generally be interpreted by readers as the reference observations. I think it's more informative to be explicit about the type of benchmark forecast used for benchmarking skill (here, you indeed choose climatology which is the most appropriate given the seasonal forecast horizon).

**2.) Pg 7 L172-173:** I'm not sure this is the correct conclusion from my interpretation of Fig. 2b and Fig. 3. It looks like skill initialised at the start of March (light green in Fig. 2b) is higher than any of the winter months, at least for the 1 week forecast horizon. This is confirmed in the map for 1 March in Fig. 3 for 1 week. Can you please clarify?

**3.)** One of the key advantages of benchmarking ESP over Sweden is the opportunity to explore the role snow accumulation and melting has on controlling ESP skill. I can't help but think there's an additional piece of the puzzle missing in attributing ESP skill. While hydrological signatures are useful, e.g. baseflow index (BFI), there is not much discussion in the paper on the hydrological processes within those catchments that are the source of ESP skill, based on information content and hence memory in the initial hydrological conditions. For example, a key question missing from the analysis is do catchments with a large contribution of streamflow from snow melt provide high skill when initialised around the snowmelt season? In practice, a catchment can have a high BFI due to several slowly responding processes (e.g. large groundwater/soil storage, snow, lakes, or a combination). I do not request this analysis is done, but it would be good to hear the authors' opinion and perhaps it could be worked into the discussion on the (initial hydrological condition) sources of ESP skill in Sweden.

**Technical comments**

**4.) Pg 3 L69:** Not sure "spread to other actors" is clear. A suggestion is: "ESP seasonal forecasts are produced operationally but have not been used widely in real-world applications due to lack of information on their skill…", or something to that effect?

**5.) Pg 4 L96-98:** Can you please confirm the timescale the KGE was calculated, I presume it was calculated at daily time step from 1981-2016?

**6.) Pg 4 Fig. 1:** Could you please add into the caption or text what exactly is shown in Fig. 1b and c in the coloured shapes, I presume it's the river network, or is it the river network downstream from an observed gauge only?

**7.) Pg 20 L365:** Suggest changing "reliable" to "skilful", as reliability was not explicitly evaluated.

**8.) Pg 21 L396-399:** "sys", "bench" and "pft" more typically subscript, not superscript (i.e. $CRPS^{sys}$ should be $CRPS_{sys}$). Also, CRPSS values can range from 1 to $-\infty$, not "low negative values".

---

## Referee Comment (RC1) · Louise Arnal (Referee) · 25 Nov 2020

In this paper, the authors evaluate the performance of an ensemble streamflow prediction (ESP) hindcast dataset for seasonal streamflow forecasting in Sweden, produced with the S-HYPE hydrological model driven by resampled historical meteorological forcings. They look at the ESP hindcast skill against a benchmark, historical streamflow climatology, for 39,493 Swedish catchments. They overall found that the ESP is skilful up to 3 months ahead in Sweden, but that the skill varies in space and time, depending on: the aggregation period selected, the catchment's hydro-climatic characteristics and regulation. They analyzed the skill against hydrological signatures, clustering basins in

7 geographical clusters in Sweden, and found that higher skill values are associated with baseflow-driven catchments.

This manuscript is overall well-written and the sound methodology leads to valuable findings both for research and for operational streamflow forecasting in Sweden. Since the focus of this manuscript is on operational forecasting to guide decision-making, further context and discussion around the potential impacts of these findings on operational decision-making is crucial. Below, please find specific comments which I hope will be helpful in shaping this manuscript further for publication.

Specific comments

Section 1:

- P1 L27: "Even if most day-to-day decisions on water-related issues are based on short- and medium-range forecasts, some activities, such as water reservoir operation and optimisation or strategic planning, benefit from long-term forecasts." Do you have any quote or public material you could share about needs of reservoir operators in Sweden? It would help emphasize the user-oriented aspect of your paper.

- P1 L29: "Despite their inherent uncertainties". I wonder if you could very briefly here cite a few examples of the uncertainties you refer to, for readers less familiar with forecasting on longer timescales?

- P2 L34: I think it is important to cite Day 1985 here (Day, G. N., 1985: Extended streamflow forecasting using NWSRFS. J. Water Resour. Plann. Manage., 111, 157–170, doi: https://doi.org/10.1061/(ASCE)0733-9496(1985)111:2(157)).

- P3 L63: "The Swedish Meteorological and Hydrological Institute (SMHI) has long been operationally providing streamflow forecasts and hydrological warnings to relevant actors in hydrological risk management (municipalities, county boards, Swedish Civil Contingencies Agency), as well as to the general public." Please clarify that this is for Sweden.

[Figure]

- P3 L69: "ESP seasonal forecasts are produced but not generally spread to other actors due to uncertainties in their skill and interpretation by external parties." This is an interesting comment and I wonder what system actors currently use for prediction on such timescales in Sweden? Please consider mentioning this in the introduction to provide some further context.

- P3 L72: "In terms of regionalisation, four main hydro-climatic regions based on hydro-climatic patterns (Lindström and Alexandersson, 2004; Pechlivanidis et al., 2018) have typically been used for water management in Sweden. However, these regions were not put forward with consideration to seasonal streamflow predictability over Sweden and might therefore be of limited use for this purpose." This appears a bit out of context here, please consider moving to the Methods section instead.

Section 2:

- P3 L86: When you say "measured values from all available stations" do you mean station observations? Please clarify here. Same for discharge and water level data. Please clarify that these are observations.

- P3 L91: Is HYPE distributed, lumped or semi-distributed? And how were the meteorological inputs prepared (e.g. interpolated) for the model to ingest?

- P3 L93: It is unclear to me at this stage how an "analysis of model outputs" was performed for 39,493 catchments if you only have 539 observation stations? Please clarify here.

- P4 L96: Please provide the lowest and highest score possible for the KGE for readers not familiar with this performance metric. Out of curiosity, has a S-HYPE model evaluation been published that you could refer readers to?

- P4 L100: I suggest putting figures 1a-c in the same order as they are mentioned in the text. I was slightly confused and thought I had missed explanations about 1b, which in fact come after 1c.

- P4 L108: "Nevertheless, since dam operation is continuously adapted (within certain bounds) to the present and most probable future meteorological and hydrological conditions, these general regulation regimes are expected to be of little benefit for seasonal forecasting purposes." This is a big statement which warrants further investigation (not necessarily in this paper though!).

- P5 L111: It may be worth explaining further how the ESP hindcasts are produced – i.e. how initial hydrological states are produced to initialize the model for each forecast start date, each meteorological forcing year corresponds to a streamflow hindcast ensemble member, etc. Perhaps a schematic would help make this clear to readers not familiar with the ESP. I also wonder what the lead time of these hindcasts is?

- P5 L129: "as a station-corrected simulation approach was used to achieve the best possible initial conditions." I am not sure to understand how a station-corrected simulation approach was used for catchments without station observations? Please clarify.

- P5 L130: Do you know if users in Sweden indeed use "ensemble forecast based on historical streamflow"?

- P6 L151: Could you please provide some more information about the k-means clustering method, or refer the readers to publicly available material further explaining this method?

Section 3.1:

- P7 L156: Please introduce Figure 2 prior to commenting on the results. What do the plots show and what is the highest/lowest score possible for the CRPSS? Same for subsequent figures.

- P7 L156: By lead time, do you mean the aggregation periods mentioned on P5 L142? Or are the results in Figure 2 from daily outputs, and up to what lead time? Please clarify here and in the Figure caption.

- P7 L162: I am not sure to understand what you mean by "the common monthly

initialisation frequency of climate prediction systems". Could you please further explain or reword?

- P7 L163: "By increasing the frequency of forecast initialisation (e.g. from once a month to once a week), and hence frequently updating the initial hydrological states, it is possible to maintain a high streamflow forecast skill for extended forecast horizons". This is a very interesting finding and I wonder if you could comment in the Discussion on how it could be translated into operational decision-making? E.g. Would decision-makers be willing to alter their decisions regularly with each forecast initialization/update?

- P8 L184: I am not sure where these lakes are in Sweden. Perhaps it would be helpful to add a map of Sweden with a few key geographical indicators (e.g. elevation, lakes – with legends for the lakes you refer to –).

- P8 L188: While I can see lower skill for the regulated rivers, it is hard to identify which rivers you refer to on L191-192. Another plot, such as a zoomed in plot, might be necessary to show these results more clearly.

- P8 L191: "future trends in streamflow". This sounds like you are looking at events (e.g. high/low flows). It is perhaps better to rephrase to "future streamflow".

- It is clever to aggregate forecasts for different periods (Figure 3). This enables to retain some skill for longer lead times than otherwise possible when looking at Figure 2. I wonder if users are interested in such time aggregations, or if they would prefer weekly/monthly aggregations instead? Could you perhaps comment on that in the Discussion, as this is important for the user-oriented analysis you are trying to achieve.

Section 3.2:

- Figure 4:

- Before looking at this figure, it wasn't clear to me that the analysis was performed for different aggregation periods as well as lead times. Could you please clarify this in the

Methods section?

- Could you please add ticks (and perhaps tick labels where possible) to all subplots of this figure as it is difficult to follow the results clearly without.

- Do you have an explanation for the sudden increase in skill for hindcasts initialized on 1 March, with a 8- vs 12-week aggregation period? Is it because you are predicting streamflow for the summer with the 12-week aggregation period, which is "easier" to predict as levels are generally low during this season? Please consider reflecting on this briefly in the paper.

- P10 L198: Could you please remind us here which aggregation periods were used for this analysis?

- P10 L203: "Even if, as expected, forecast skill decreases when forecasts are aggregated over long periods, a comparatively higher skill is maintained over longer time horizons than when forecasts are aggregated over short periods." It would be interesting if you could add an indication of the lead time at which the skill is 0 for shorter aggregation periods (results from Figure 2) on this figure.

Section 3.3:

- I would argue that results for longer forecast horizons would be good to show as well as the focus of this paper is on seasonal forecasting. Perhaps correlations could be stronger when calculated against another performance metric which might not weaken so much over time (e.g. CRPS instead of its skill score)?

- To what extent do you think these results are dependent on your hydrological model? Please consider commenting on this in the Discussion section.

- Could you please increase the font size of the correlation coefficient on each subplot of Figure 5? It took me a bit of time to notice them.

Section 3.4:

- Table 2: It would be good to show the range of elevation, annual precipitation, etc. instead of just the mean values, to show the catchments variability within each cluster region. This might become a bit messy and could be clearer in a figure rather than a table.

- P14 L241-254: It may be easier to follow by having these observations as bullet points in Table 2. It might also make it easier to link the results presented in Figure 7 with the cluster characteristics.

- Could the large/small spread in forecast skill shown in Figure 7 be caused by large/small basin differences within these clusters? E.g. spread in the topographic, climatological or hydrological characteristics (from Table 2) within each cluster. It would be interesting to hear your thoughts on this here on in the Discussion. For example, cluster 5 catchments appear more spread out throughout Sweden (Figure 6b) compared to cluster 6 catchments.

Section 4:

- P18 L306: "forecast initialisations are not expected to provide an added value to the forecast service." I would argue the opposite. You have shown in your paper that more frequent forecast initializations could substantially increase the forecast skill. The added value is potentially immense for decision-makers. The challenge remains to translate this into actionable outputs for the users, as you mention it briefly. Please consider rephrasing and elaborating on this.

- P19 L332: Would you be able to add a figure to the paper to support these very interesting findings?

- P19 L344: "Skilful ESP seasonal forecasts for these rivers should allow for early planning and allocation of resources that could greatly contribute to mitigate potentially severe ice break-ups." To evaluate this, a different performance metric, such as the brier or ROC score for high flow events, might be better adapted than the CRPS. Do

you plan to look at this in the future?

---

## Referee Comment (RC2) · Anonymous Referee #2 · 27 Nov 2020

This manuscript presents a large scale study (39 493 catchments) that aims at gaining a better understanding of the main factors that drive the skill of ensemble streamflow forecasts in Sweden. Most similar studies in seasonal forecasting aim at distinguishing the contribution of initial conditions and that of meteorological forcings. In this manuscript, the authors rather want to distinguish the hydrological processes that drive the skill of seasonal forecasts across space and time. They also study the influence of aggregating the forecasts at different time scale (2 weeks, 1 months, 2 months, etc.) affects the skill, which I find very interesting. The authors show that forecasts are mostly skillful when initialized during the winter months, and for base flow dominated catchments. They also propose a classification of catchments into clusters with similar

characteristics and behavior relative to seasonal forecasts. I think this is an interesting study, that can bring new information to better understand where we should concentrate our efforts to improve the skill of seasonal forecasts in hydrology. I only have very minor comments, that relate to methodological choices that I would like the authors to explain in greater details.

*Detailed comments:*

- Line 34: I am always bothered when people change the original name of a technique. The authors here define ESP as "Ensemble Streamflow Predictions", but this is not exactly what ESP originally stand for. In Day (1985), who originally proposed the technique, ESP refers to "Extended Streamflow Prediction". This may seem like a small detail, but 1) I think it is only fair to use the exact name that Day proposed for his own technique and 2) "Ensemble" prediction is very general and could very well be obtained using a dynamical meteorological model rather than past climatological scenarios. Therefore, designating ESP as "Ensemble" streamflow prediction can be confusing to some readers (I'm thinking especially about people who are unfamiliar with ensemble forecasting in general). ESP should refer to a very specific technique, but I have also heard people using it to refer to ensemble forecasts obtained using dynamical meteorological forecasts. Also, I think that Day (1985) should be cited, as it is the original reference for ESP.

- Lines 34-50 and lines 291-299: Speaking of dynamical forecasts: ESP is quite an old technique. And I agree that it is still what is used operationally for long-term hydrological forecasting by many operational agencies, and that it works well. However, long-term dynamical meteorological forecasts also exist and some studies focus on assessing their skill for hydrology, often using ESP as a reference for comparison (e.g. Meißner et al. 2017; Baker et al. 2019, Slater et al. 2019, Bazile et al. 2017 and others). I don't have any problem with the authors using ESP instead of dynamical forecasts, but I think the use of dynamical me-

teorological for seasonal hydrological forecasting should also be included in the literature review. There is a good discussion about NWP later in the paper (291-299), but I think it appears much too late. I strongly suggest including examples of NWP-based hydrological seasonal forecasting systems in the introduction, and possibly moving some elements from the discussion (a portion of lines 291-299) also in the introduction. I think it is important to explain why you chose to use ESP rather than NWP based forecasts, and to do so before the discussion!

- Page 4 lines 101-110: I'm not sure I understand why it is relevant to include regulated rivers in the study. They all end up in the same cluster (7), which unsurprisingly has a negative median skill. It would certainly be interesting to forecasts long-term inflows to reservoirs, as it could be useful for long term water management/hydropower production planning, but if I understand those lines correctly, this doesn't seem to be the case here (I understand that there are forecast points downstream from reservoirs, correct?). I think the rationale for including regulated catchment in the study needs to be better explained.

- Page 14 lines 240-253 and Figure 6: I would find it helpful if the abbreviations from Table 1 were used in this paragraph, which analyses Figure 6 (even though a sort of synthesis is presented in Table 2). I find it difficult to remember acronyms and abbreviations, so I had to go back and for the between the figure, the text and Table 1.

- Table 2: How are potential and actual evapotranspiration obtained? Is it really important to include both in the table?

- Page 16 line 268: Do you have any possible explanation why the cluster (5) with the highest general skill also have the largest spread? Is it possible that those two things (skill and spread) are related? What I mean is that if the skill is assessed by the CRPS and the CRPS is very sensitive to spread, then maybe the high skill

is (at least in part) a consequence of this high spread? In any case, I think it would be interesting if the authors could provide a possible explanation.

- Page 18 lines 316-326: You mention the idea of using more sophisticated data assimilation techniques, such as the EnKF, but I think it would also be worth mentionning the possibility of assimilating other observations than streamflow, for instance soil moisture and/or snow water equivalent. This has been done in some studies (e.g. Huan et al. 2017), but there are still not that many in direct relation to seasonal forecasting.

- Page 19 lines 335-337: "This exercise shows that the regulation routines in ..." There I finally found the justification for including regulated rivers in the study. I think this should be expressed earlier in the manuscript, around lines 105-120. At the moment, the explanations provided in lines 105-120 remain too general and it is hard to understand what it is that you want to test by including regulated rivers. At lines 335-337 it becomes clear, but it is too late.

References:

Day, G. (1985). Extended Streamflow Forecasting Using NWSRFS, J. Wat. Res. Plan. Mgmt., 10.1061/(ASCE)0733-9496(1985)111:2(157), 157-170

MeiBner et al. 2017 (already cited in the manuscript)

Baker S.A., Wood A.W. and Rajagopalan B. (2019). Developing Subseasonal to Seasonal Climate Forecast Products for Hydrology and Water Management, Journal of the American Water Resources Association, 55(4), 1024-1037.

Slater L.J., Villarini G., Bradley A.A. and Vecchi G.A. (2019) A dynamical statistical framework for seasonal forecasting in an agricultural watershed, Climate Dynamics, 53(12), 7429-7445.

Bazile R., Boucher M-A, Perreault L. And Leconte R. (2017) Verification of ECMWF

System 4 for seasonal hydrological forecasting in a northern climate, Hydrology and Earth System Sciences, 21, 5747–5762.

Huang C., Newman A.J., Clark M.P., Wood A.W. and Zheng X. (2017) Evaluation of snow data assimilation using the ensemble Kalman filter for seasonal streamflow prediction in the western United States, Hydrology and Earth System Sciences, 21, 635–650.

---

## Author Comment (AC1) · 18 Dec 2020

**Authors' response to interactive comment by Dr Louise Arnal**

Black text: Reviewer comment

Blue text: Authors' response

In this paper, the authors evaluate the performance of an ensemble streamflow prediction (ESP) hindcast dataset for seasonal streamflow forecasting in Sweden, produced with the S-HYPE hydrological model driven by resampled historical meteorological forcings. They look at the ESP hindcast skill against a benchmark, historical streamflow climatology, for 39,493 Swedish catchments. They overall found that the ESP is skilful up to 3 months ahead in Sweden, but that the skill varies in space and time, depending on: the aggregation period selected, the catchment's hydro-climatic characteristics and regulation. They analyzed the skill against hydrological signatures, clustering basins in 7 geographical clusters in Sweden, and found that higher skill values are associated with baseflow-driven catchments. This manuscript is overall well-written and the sound methodology leads to valuable findings both for research and for operational streamflow forecasting in Sweden. Since the focus of this manuscript is on operational forecasting to guide decision-making, further context and discussion around the potential impacts of these findings on operational decision-making is crucial. Below, please find specific comments which I hope will be helpful in shaping this manuscript further for publication.

We thank Dr Louise Arnal for her valuable comments and suggestions that will undoubtedly help us improve our manuscript. Below we reply to each of the comments and explain how we will incorporate them into the manuscript.

Specific comments

Section 1:

- P1 L27: "Even if most day-to-day decisions on water-related issues are based on short- and medium-range forecasts, some activities, such as water reservoir operation and optimisation or strategic planning, benefit from long-term forecasts." Do you have any quote or public material you could share about needs of reservoir operators in Sweden? It would help emphasize the user-oriented aspect of your paper.

Here we plan to reference the work by Foster et al. (2018), which refers to the Swedish hydropower needs. In addition, the recently accepted publication by Giuliani et al. (2020) quantifies the added economic value from incorporating seasonal forecasts in a regulated reservoir for the agriculture sector as well as for flood prevention. Finally, the public deliverable D2.2 from the S2S4E project (S2S4E, 2018) highlights the user needs from various users in the energy sector.

- P1 L29: "Despite their inherent uncertainties". I wonder if you could very briefly here cite a few examples of the uncertainties you refer to, for readers less familiar with forecasting on longer timescales?

We will include some examples of these uncertainties, such as hydro-meteorological model errors, future atmospheric states and past hydro-meteorological water storage, in the revised version of the manuscript.

- P2 L34: I think it is important to cite Day 1985 here (Day, G. N., 1985: Extended streamflow forecasting using NWSRFS. J. Water Resour. Plann. Manage., 111, 157–170, doi: https://doi.org/10.1061/(ASCE)0733-9496(1985)111:2(157)).

We will include this reference in the revised version of the manuscript.

- P3 L63: "The Swedish Meteorological and Hydrological Institute (SMHI) has long been operationally providing streamflow forecasts and hydrological warnings to relevant actors in hydrological risk management (municipalities, county boards, Swedish Civil Contingencies Agency), as well as to the general public." Please clarify that this is for Sweden.

We will clarify this in the revised version of the manuscript.

- P3 L69: "ESP seasonal forecasts are produced but not generally spread to other actors due to uncertainties in their skill and interpretation by external parties." This is an interesting comment and I wonder what system actors currently use for prediction on such timescales in Sweden? Please consider mentioning this in the introduction to provide some further context.

Both actors and the general public have access to the current hydrological situation and streamflow climatology through the open access Vattenwebb portal (available at https://vattenwebb.smhi.se), which they can use to get information on the latest observed streamflow values as well as to get an estimate of the most likely discharge for any given season based on historical discharges. On top of that, SMHI's consultancy services provide tailored forecasts to relevant actors. These forecasts are however not included in the public service and, as of today, are limited to individual river systems. We will include this information in the revised manuscript.

- P3 L72: "In terms of regionalisation, four main hydro-climatic regions based on hydro-climatic patterns (Lindström and Alexandersson, 2004; Pechlivanidis et al., 2018) have typically been used for water management in Sweden. However, these regions were not put forward with consideration to seasonal streamflow predictability over Sweden and might therefore be of limited use for this purpose." This appears a bit out of context here, please consider moving to the Methods section instead.

The original thought was to present this as background information to the clustering analysis, hence its placement in the introduction. However, we agree with the reviewer in that it would fit better in the methods section. We will move it there in the revised version of the manuscript.

Section 2:

- P3 L86: When you say "measured values from all available stations" do you mean station observations? Please clarify here. Same for discharge and water level data. Please clarify that these are observations.

The reviewer is correct, we refer to observations here. We will revise the manuscript to ensure that the correct term is used throughout the text.

- P3 L91: Is HYPE distributed, lumped or semi-distributed? And how were the meteorological inputs prepared (e.g. interpolated) for the model to ingest?

We note here the HYPE refers to the model, while S-HYPE refers to the Swedish implementation of the HYPE model. In previous investigations, the HYPE model has been used in lumped, semi-distributed and distributed modes. That being said, the S-HYPE model setup is semi-distributed, so gridded meteorological inputs need to be averaged for each model catchment in order to be used. In this case, we follow the same methodology as in the operational service. This way, the meteorological inputs are processed using a weighted average method based on the area fraction of a given S-HYPE catchment covered by each cell of the gridded dataset (only cells which partially or totally overlap the area of the given catchment are assigned weights). We will clarify this in the revised version of the manuscript.

- P3 L93: It is unclear to me at this stage how an "analysis of model outputs" was performed for 39,493 catchments if you only have 539 observation stations? Please clarify here.

The reference used in the evaluation is based on a combination of observations and perfect forecasts, and can therefore cover all 39,493 catchments. This hybrid reference was chosen because it corresponds to SMHI's operational setup. This is actually a common setup of operational services, which includes assimilation of available observations in order to improve the representation of local initial conditions. Therefore, the analysis is performed on all 39,493 catchments, most of them being analysed against perfect forecasts. For those catchments associated with one of the 539 observation stations, the model outputs are instead assessed against those observations (the model outputs themselves are corrected with existing observations before initialising the forecasts and AR-updating is used when observations are no longer available). Catchments located downstream observation stations partially benefit from the model corrections made upstream, and the reference thus becomes a mix of observed discharges flowing into downstream modelled catchments. We will clarify this in the revised version of the manuscript.

- P4 L96: Please provide the lowest and highest score possible for the KGE for readers not familiar with this performance metric. Out of curiosity, has a S-HYPE model evaluation been published that you could refer readers to?

We will add the KGE ranges in the revised manuscript, as suggested by the reviewer. In this investigation we used the S-HYPE 2016 version which is the latest operational model version. Since the S-HYPE model is subject to continuous efforts, the performance of the current version of the model performance has not yet been published; hence we are here firstly reporting the evaluation results.

- P4 L100: I suggest putting figures 1a-c in the same order as they are mentioned in the text. I was slightly confused and thought I had missed explanations about 1b, which in fact come after 1c.

We will modify the order of the subplots in Fig 1. following the suggestion from the reviewer.

- P4 L108: "Nevertheless, since dam operation is continuously adapted (within certain bounds) to the present and most probable future meteorological and hydrological conditions, these general regulation regimes are expected to be of little benefit for seasonal forecasting purposes." This is a big statement which warrants further investigation (not necessarily in this paper though!).

What we tried to convey here is that, since dam operation needs to be continuously adjusted to the changing hydro-meteorological conditions, in addition to consider other factors such as optimising the economic benefit and ensuring safe operation, long-range hydrological forecasts based on models with only a limited description of such complex decisions on regulation patterns will most likely be conditioned by these simplifications. We agree with the reviewer that further investigation would be needed to justify a clear statement on this. We will clarify this in the revised version of the manuscript to avoid any misunderstandings on this matter.

- P5 L111: It may be worth explaining further how the ESP hindcasts are produced – i.e. how initial hydrological states are produced to initialize the model for each forecast start date, each meteorological forcing year corresponds to a streamflow hindcast ensemble member, etc. Perhaps a schematic would help make this clear to readers not familiar with the ESP. I also wonder what the lead time of these hindcasts is?

We will include a schematic of how ESP hindcasts and benchmark forecasts are produced in the revised manuscript. Regarding the lead time of the hindcasts, we used 190 days (~6 months). We will specify this in the revised manuscript.

- P5 L129: "as a station-corrected simulation approach was used to achieve the best possible initial conditions." I am not sure to understand how a station-corrected simulation approach was used for catchments without station observations? Please clarify.

This was, of course, only possible for catchments where observations were available. Nevertheless, even catchments downstream from observations were partially benefited from this station-correction approach. Elsewhere, model outputs were simply simulation results. We will clarify this in the revised manuscript. Additionally, we will also include this step (i.e. station correction) in the new schematic showing ensemble generation (see previous comment).

- P5 L130: Do you know if users in Sweden indeed use "ensemble forecast based on historical streamflow"?

As explained in an earlier comment, this information, together with the latest observations, is openly available through SMHI's Vattenwebb portal (available at https://vattenwebb.smhi.se). The general public and other actors are encouraged to use this information to (i) get an estimation of expected discharge at any given season and, (ii) to see whether the latest observations are lower or higher than normal. That being said, it is difficult to quantify the actual use of ensemble forecasts in Sweden. From general discussions, we can say that advanced users from the hydropower sector are

used to ensemble forecasts based on historical streamflow (some of them also use forecasts based on ESP and NWP techniques), while other sectors may be more familiar with deterministic information. Other tailored SMHI services using ensemble forecasts based on historical records such as the Aqua service (https://europa.eu/!bB63kr) are set up for the water supply authorities.

- P6 L151: Could you please provide some more information about the k-means clustering method, or refer the readers to publicly available material further explaining this method?

In the revised manuscript we will refer to Jin & Han (2011), which nicely summarizes the concept of k-means clustering.

Section 3.1:

- P7 L156: Please introduce Figure 2 prior to commenting on the results. What do the plots show and what is the highest/lowest score possible for the CRPSS? Same for subsequent figures.

Here we think that the figures are adequately introduced in the captions and that, therefore, including an additional introduction in the main text would lead only to redundant information in the manuscript. Nevertheless, in the revised version of the manuscript we will make an additional effort to ensure that the necessary information for understanding all figures (e.g. highest/lowest possible CRPSS) is available to the reader in an intuitive way.

- P7 L156: By lead time, do you mean the aggregation periods mentioned on P5 L142? Or are the results in Figure 2 from daily outputs, and up to what lead time? Please clarify here and in the Figure caption.

We produced daily forecasts up to 190 days into the future and then calculated weekly averages. So, on Figure 2a, "0 Mn" refers to the first forecast week, "1 Mn" to the fifth forecast week, and so on. The aggregation periods mentioned on P5 L142 refer to Section 3.2. In the revised manuscript we will clarify this both in the text and in Figure 2. Note that we are using "lead time" and "forecast time" definitions from the Copernicus Climate Change Service, i.e. for weekly aggregations, lead week 0 is the same as forecast week 1.

- P7 L162: I am not sure to understand what you mean by "the common monthly initialisation frequency of climate prediction systems". Could you please further explain or reword?

By this we meant that, even though we now see more frequent forecast initialisations in some systems, many seasonal climate forecasts are initialised and produced once a month. We will rephrase this in the revised version of the manuscript.

- P7 L163: "By increasing the frequency of forecast initialisation (e.g. from once a month to once a week), and hence frequently updating the initial hydrological states, it is possible to maintain a high streamflow forecast skill for extended forecast horizons". This is a very interesting finding and I wonder if you could comment in the Discussion on how it could be translated into operational decision-making? E.g. Would decision-makers be willing to alter their decisions regularly with each forecast initialization/update?

This is a good point which we plan to address in the Discussion section of the revised manuscript.

Here we state that the way a seasonal forecasting service is used in decision-making depends on the sector, user, and service properties. It is therefore important to evaluate a comprehensive range of possibilities in terms of seasonal information statistics (e.g. forecast aggregation, time horizons) that can technically be offered to individual decision-makers to allow flexibility in the decision process. It is also important to point out that here we can only hypothesize on the impacts of our findings on decision contexts, which are very much sector and location-dependent.

Our findings show that a frequent (i.e. weekly) initialisation can significantly improve the streamflow forecast skill, and this is expected to add value to decision-making. This is of particular high importance for periods in which decisions are subjected to hydrological responses that alter in a short time window. For instance, in Sweden it is important to be able to predict the onset of the spring flood due to a combination of snow melting and precipitation, and adjust the reservoir regulation accordingly to optimize the power production for the coming months.

- P8 L184: I am not sure where these lakes are in Sweden. Perhaps it would be helpful to add a map of Sweden with a few key geographical indicators (e.g. elevation, lakes – with legends for the lakes you refer to –).

In the revised version of the manuscript we will include an additional figure in the appending in which we will show the elevation, and hydrography of Sweden. Additionally, we will locate the main Swedish rivers and lakes that are named throughout the manuscript in this new figure as well.

- P8 L188: While I can see lower skill for the regulated rivers, it is hard to identify which rivers you refer to on L191-192. Another plot, such as a zoomed in plot, might be necessary to show these results more clearly.

As mentioned in the previous comment, we plan to include a figure with the main Swedish rivers and lakes. This would allow a clearer identification of the river systems we refer to.

- P8 L191: "future trends in streamflow". This sounds like you are looking at events (e.g. high/low flows). It is perhaps better to rephrase to "future streamflow".

We will reformulate this following the reviewer's suggestion, as it may indeed be clearer for readers.

- It is clever to aggregate forecasts for different periods (Figure 3). This enables to retain some skill for longer lead times than otherwise possible when looking at Figure 2. I wonder if users are interested in such time aggregations, or if they would prefer weekly/monthly aggregations instead? Could you perhaps comment on that in the Discussion, as this is important for the user-oriented analysis you are trying to achieve.

As mentioned earlier, the temporal aggregations depend on the sector and user. For instance, for the energy sector, the hydropower companies tend to be interested in a fixed 3-month aggregation over the period May-July. Alternatively, crop water needs can be assessed over the entire summer season to get estimates of required water volumes for irrigation. The produced matrix (Figure 4) for different aggregations, initializations, and lead times allows communication of skill to various users

depending on their needs. We will include these considerations in the revised version of the manuscript.

Section 3.2:

- Figure 4:

- Before looking at this figure, it wasn't clear to me that the analysis was performed for different aggregation periods as well as lead times. Could you please clarify this in the Methods section?

We explained briefly the analysis using different aggregation periods in P5 L139-143. In the revised manuscript we will reformulate this so it is clearer for the reader that we also perform this type of analysis.

- Could you please add ticks (and perhaps tick labels where possible) to all subplots of this figure as it is difficult to follow the results clearly without.

As suggested by the reviewer, we will add ticks for all subplots in this figure in the revised manuscript. Additionally, we will add labels to the y-axes of the subplots for January, April, July, and October, and to the x-axes of the subplots for October, November, and December.

- Do you have an explanation for the sudden increase in skill for hindcasts initialized on 1 March, with a 8- vs 12-week aggregation period? Is it because you are predicting streamflow for the summer with the 12-week aggregation period, which is "easier" to predict as levels are generally low during this season? Please consider reflecting on this briefly in the paper.

This increase in skill, which is particularly obvious in March, can in fact be observed for hindcasts initialised between 1 February to 1 May when looking at the 4-week aggregation period, and corresponds roughly to the month of May. Many catchments and rivers, especially in the northern half of the country, see the peak of the spring flood during this month. With shorter aggregation periods, the focus is more influenced on the start/end of the event, while longer aggregations put more emphasis on having a correct total volume, regardless of the exact start/end dates. Since this total volume linked to the accumulated snowpack is easier to model than the timing of the event, which is conditioned by meteorological variables, longer aggregations perform better. In the southern parts of the country, in the month of May the spring flood has already passed and low flow conditions start to dominate. We will include these considerations in the revised version of the manuscript.

- P10 L198: Could you please remind us here which aggregation periods were used for this analysis?

We will follow the reviewer's suggestion and add the aggregation periods we used in the analysis here.

- P10 L203: "Even if, as expected, forecast skill decreases when forecasts are aggregated over long periods, a comparatively higher skill is maintained over longer time horizons than when forecasts are aggregated over short periods." It would be interesting if you could add an indication of the lead time at which the skill is 0 for shorter aggregation periods (results from Figure 2) on this figure.

The bottom row of each subplot in Figure 4 contains already the same information as Figure 2b, as the aggregation period (i.e. 1 week) is exactly the same. So, the first grey box in the bottom line of each subplot already shows this information. So, after discussion with the co-authors, we decided to avoid making this figure heavier than it is, as the objective here is to depict how the skill changes as a function of the aggregation window, and not only when this drops below 0.

Section 3.3:

- I would argue that results for longer forecast horizons would be good to show as well as the focus of this paper is on seasonal forecasting. Perhaps correlations could be stronger when calculated against another performance metric which might not weaken so much over time (e.g. CRPS instead of its skill score)?

The results presented in Section 3.3 correspond to an exploratory investigation connecting the first part of the analysis (i.e. temporal and spatial variability of ESP forecast skill) with the second part (i.e. attribution of skill to hydrological behaviour). By focusing on the CRPSS, we look at the "added value" of the ESP with respect to streamflow climatology, which is in line with the idea of evaluating/understanding the use of ESP for decision making (against an alternative system). Looking at the CRPS or any score without a benchmark would be a different analysis completely which would undoubtedly be very interesting but which is outside the scope of this study. That being said, in the revised version of the manuscript we will address this comment by adding the results for a further lead time in light grey in the same figure.

- To what extent do you think these results are dependent on your hydrological model? Please consider commenting on this in the Discussion section.

Different aspects of the S-HYPE modelling and forecasting chain in this study, such as the model setup and data, the model structure, and its parameters may convey uncertainty to the forecast results (see also the discussion in Pechlivanidis et al., 2020). However, the impact of model errors for our particular setup is especially complex as we used a combination of observations and perfect forecasts as reference. While we can expect model errors to be minimal for those catchments in which forecasts are purely evaluated against perfect forecasts, they become relevant for catchments at or downstream of observations, especially due to the interplay between correction of model outputs with observations and streamflow regulation.

While model outputs are corrected with all available observations, not all watercourses with observations are regulated, and even those that are regulated do not have all observations at dams or other river regulation structures. The correction of model outputs with observations and, when these are no longer available (e.g. at forecast initialization), with an exponentially decreasing factor based on the last known model error (i.e. AR correction) effectively minimises model uncertainties, especially at forecast initialisation and during the first time steps of the forecast. Nevertheless, any model errors will tend to become more significant for further lead times. The downstream distance of a given catchment with respect to an observation is also relevant in this case, as the model correction would only affect a fraction of the simulated/forecasted streamflow at that location.

The most important model errors, though, can be expected for heavy regulated catchments with or downstream of observations. Complex river regulation routines which depend on factors external to hydrological models cannot be adequately reproduced by these models. In these cases, even if the correction of model outputs with observations may minimise model errors at forecast initialisation, these errors will rapidly spread due to the inability of the model to reproduce the modified hydrological regime.

We will include these considerations in the revised version of the manuscript.

- Could you please increase the font size of the correlation coefficient on each subplot of Figure 5? It took me a bit of time to notice them.

Following the reviewer's advice, we will increase the font size of the text of Figure 5 to make it more readable.

Section 3.4:

- Table 2: It would be good to show the range of elevation, annual precipitation, etc. instead of just the mean values, to show the catchments variability within each cluster region. This might become a bit messy and could be clearer in a figure rather than a table.

In the revised version of the manuscript we will include the interquartile ranges (Q25 - Q75) in addition to the mean values for each of the variables. Following a comment by another reviewer, we will remove potential evapotranspiration from the table, which will give more space for the additional information.

- P14 L241-254: It may be easier to follow by having these observations as bullet points in Table 2. It might also make it easier to link the results presented in Figure 7 with the cluster characteristics.

The text in L241-254 refers to the dominant hydrological processes and topographic characteristics, while Table 2 summarizes the streamflow signatures which define the clusters. We propose not to add similar information in Table 2 and hence introduce redundancy in the manuscript. We will nevertheless make an effort to make this paragraph easier to follow by the reader in the revised version of the manuscript.

- Could the large/small spread in forecast skill shown in Figure 7 be caused by large/small basin differences within these clusters? E.g. spread in the topographic, climatological or hydrological characteristics (from Table 2) within each cluster. It would be interesting to hear your thoughts on this here on in the Discussion. For example, cluster 5 catchments appear more spread out throughout Sweden (Figure 6b) compared to cluster 6 catchments.

The hydrological characteristics are the end-product of climatological and physiographic properties and can therefore not be assessed together. Some combinations of climatological and physiographic properties can be found in very specific areas of the country, while others are more widespread. For instance, from a physiographic perspective, cluster 6 consists mainly of agricultural and coastal catchments, in addition to big lakes, which are quite limited geographically in Sweden. Conversely,

cluster 5 contains mostly slowly-responding forested catchments, which can be found throughout the country.

Focusing on the hydrological characteristics, results from cluster 5 are indeed interesting. The forecasts in the catchments clustered here generally show the highest skill (for all lead times) among all cluster groups, yet results are widely spread. In this paper we conclude that forecast skill is strongly linked to the various hydrological regimes (see also a more detailed investigation in Pechlivanidis et al. 2020), and hence we argue that the reason for this spread lies in a deeper understanding of the hydrological signatures in cluster 5. As we state in P14 L241-242, the catchments in cluster 5 are characterized by a high baseflow contribution (BFI), a slow response to precipitation (Flash) and a generally small intra-annual variability (DPar). In Figure 6a we observe that, although the mean values for RLD (rising limb density) are below the 33rd percentile of this signature (which represent 'below normal' signature values), the variability among the 4355 catchments composing this cluster is high (as indicated by the boxplot), with some catchments experiencing 'normal' RLD values and yet some others with values even higher than the 66th percentile of this signature. Consequently, this indicates that, despite their high baseflow contribution, some catchments in cluster 5 experience sharp increases in their hydrographs, which is an indication of low skill as seen in Figure 5 (CRPSS and RLD are strongly, but negatively, correlated). We will explain the above argument for the large spread in cluster 5 in the revised manuscript.

Section 4:

- P18 L306: "forecast initialisations are not expected to provide an added value to the forecast service." I would argue the opposite. You have shown in your paper that more frequent forecast initializations could substantially increase the forecast skill. The added value is potentially immense for decision-makers. The challenge remains to translate this into actionable outputs for the users, as you mention it briefly. Please consider rephrasing and elaborating on this.

Frequent initialization as seen in this manuscript (i.e. weekly with respect to monthly), does provide added skill. However, we argue that daily initialization (when compared to weekly initialization) is unlikely to convey any further useful information for decision making at seasonal horizons, since long-term decisions are also not taken daily. In such services, due to high uncertainty, results are aggregated into weekly values, which further smooth the potentially high streamflow dynamics. We will clarify this in the revised version of the manuscript.

- P19 L332: Would you be able to add a figure to the paper to support these very interesting findings?

Here we want to clarify that the sentence in P19 L332 does not correspond to actual findings presented in this manuscript, which build on an analysis of the operational forecasting setup from the perspective of public service, thus focusing on catchment outflows. Instead, this statement is based on the assumption that, since forecast skill is shown to be consistently lower in highly regulated catchments than elsewhere, the fraction of the inflows to a given reservoir that are not affected by other regulation upstream may be more predictable and therefore convey higher forecast skill when compared to the outflows, which would be very relevant for the hydropower

sector. This is indeed a very interesting analysis that we plan to investigate further in the future. We will clarify this in the revised version of the manuscript.

- P19 L344: "Skilful ESP seasonal forecasts for these rivers should allow for early planning and allocation of resources that could greatly contribute to mitigate potentially severe ice break-ups." To evaluate this, a different performance metric, such as the brier or ROC score for high flow events, might be better adapted than the CRPS. Do you plan to look at this in the future?

The severity of ice break-ups is determined by the interplay of different factors and processes over a long period, usually starting in late autumn. The main drivers are meteorological (defining the ice build-up during the winter months and meltdown during spring) and hydrological (regarding the timing of the streamflow increase marking the start of the spring flood). So, here we argue that scores which evaluate the overall performance, including biases in volume, such as the CRPS are also suitable for decision-making on the allocation of resources. That being said, we do plan to explore this further into the future, including looking at the metrics suggested by the reviewer.

**References**

Foster, K., C. B. Uvo, and J. Olsson (2018), The development and evaluation of a hydrological seasonal forecast system prototype for predicting spring flood volumes in Swedish rivers, Hydrol. Earth Syst. Sci., 22(5), 2953–2970, doi:10.5194/hess-22-2953-2018.

Giuliani, M., Crochemore, L., Pechlivanidis, I., and Castelletti, A.: From skill to value: isolating the influence of end-user behaviour on seasonal forecast assessment, Hydrol. Earth Syst. Sci. Discuss., https://doi.org/10.5194/hess-2019-659, accepted, 2020.

Jin X., Han J. (2011) K-Means Clustering. In: Sammut C., Webb G.I. (eds) Encyclopedia of Machine Learning. Springer, Boston, MA. https://doi.org/10.1007/978-0-387-30164-8_425

Pechlivanidis, I. G., Crochemore, L., Rosberg, J. and Bosshard, T. (2020): What Are the Key Drivers Controlling the Quality of Seasonal Streamflow Forecasts?, Water Resour. Res., 56(6), doi:10.1029/2019WR026987.

S2S4E (2018) Deliverable 2.2 User needs and decision-making processes that can benefit from S2S forecasts.

---

## Author Comment (AC2) · 18 Dec 2020

**Authors' response to interactive comment by Anonymous Reviewer #2**

Black text: Reviewer comment

Blue text: Authors' response

This manuscript presents a large scale study (39 493 catchments) that aims at gaining a better understanding of the main factors that drive the skill of ensemble streamflow forecasts in Sweden. Most similar studies in seasonal forecasting aim at distinguishing the contribution of initial conditions and that of meteorological forcings. In this manuscript, the authors rather want to distinguish the hydrological processes that drive the skill of seasonal forecasts across space and time. They also study the influence of aggregating the forecasts at different timescale (2 weeks, 1 months, 2 months, etc.) affects the skill, which I find very interesting. The authors show that forecasts are mostly skillful when initialized during the winter months, and for base flow dominated catchments. They also propose a classification of catchments into clusters with similar characteristics and behavior relative to seasonal forecasts. I think this is an interesting study that can bring new information to better understand where we should concentrate our efforts to improve the skill of seasonal forecasts in hydrology. I only have very minor comments that relate to methodological choices that I would like the authors to explain in greater detail.

We thank the reviewer for his/her valuable comments and suggestions that will undoubtedly help us improve our manuscript. Below we reply to each of these and explain how we will incorporate them into the manuscript.

Detailed comments:

• Line 34: I am always bothered when people change the original name of a technique. The authors here define ESP as "Ensemble Streamflow Predictions", but this is not exactly what ESP originally stands for. In Day (1985), who originally proposed the technique, ESP refers to "Extended Streamflow Prediction". This may seem like a small detail, but 1) I think it is only fair to use the exact name that Day proposed for his own technique and 2) "Ensemble" prediction is very general and could very well be obtained using a dynamical meteorological model rather than past climatological scenarios. Therefore, designating ESP as "Ensemble" streamflow prediction can be confusing to some readers (I'm thinking especially about people who are unfamiliar with ensemble forecasting in general). ESP should refer to a very specific technique, but I have also heard people using it to refer to ensemble forecasts obtained using dynamical meteorological forecasts. Also, I think that Day (1985) should be cited, as it is the original reference for ESP.

We agree with the reviewer in that terminology should be used in a restrictive sense and that original ideas and their naming should be respected and used. That being said, the term Ensemble Streamflow Prediction referring to Day's 1985 technique has been widely adopted by the community and has nowadays almost replaced the original term (Extended Streamflow Prediction), which is why

we use it even here. Nonetheless, the reviewer makes a good point here, and following his/her reflection we will also include a reference to the original name of the ESP technique and publication in the revised manuscript.

• Lines 34-50 and lines 291-299: Speaking of dynamical forecasts: ESP is quite an old technique. And I agree that it is still what is used operationally for long-term hydrological forecasting by many operational agencies, and that it works well. However, long-term dynamical meteorological forecasts also exist and some studies focus on assessing their skill for hydrology, often using ESP as a reference for comparison (e.g. Meißner et al. 2017; Baker et al. 2019, Slater et al. 2019, Bazile et al. 2017 and others). I don't have any problem with the authors using ESP instead of dynamical forecasts, but I think the use of dynamical meteorological for seasonal hydrological forecasting should also be included in the literature review. There is a good discussion about NWP later in the paper (291-299), but I think it appears much too late. I strongly suggest including examples of NWP-based hydrological seasonal forecasting systems in the introduction, and possibly moving some elements from the discussion (a portion of lines 291-299) also in the introduction. I think it is important to explain why you chose to use ESP rather than NWP based forecasts, and to do so before the discussion!

In the revised manuscript, we will include a short description of NWP-based techniques in the introduction and further clarify the reasons behind the choice of ESP in this study. These reasons include the fact that the objective of this manuscript was to assess the existing system at SMHI's operational service, which uses ESP forecasts, and that the ESP method offers the best study object to focus on the role of initial hydrologic conditions alone (best explained through catchment characteristics than the role NWP forcings).

• Page 4 lines 101-110: I'm not sure I understand why it is relevant to include regulated rivers in the study. They all end up in the same cluster (7), which unsurprisingly has a negative median skill. It would certainly be interesting to forecasts long-term inflows to reservoirs, as it could be useful for long term water management/hydropower production planning, but if I understand those lines correctly, this doesn't seem to be the case here (I understand that there are forecast points downstream from reservoirs, correct?). I think the rationale for including regulated catchment in the study needs to be better explained.

In our view, a clear explanation to this is provided in the discussion section of the manuscript, as the reviewer states in a later comment. The rationale behind this is an operational one: since the operational service we are trying to evaluate here includes regulated rivers (which are, additionally, of special interest for such a system), they should be taken into the account in the analysis as well. It should be noted though, that the degree of regulation is not explicitly considered as one of the indicators for the clustering analysis. Nevertheless, since the regulation scheme affects the hydrological response, it is plausible that regulated catchments become clustered together.

Regarding the evaluation of inflows to reservoirs, we agree with the reviewer in that this would be very relevant for long term water management and hydropower production planning. However, in this manuscript we focused on the operational forecasting setup from the perspective of public service, which provides information based on catchment outflows. Nevertheless, even if this analysis

is out of the scope of the present manuscript, it is something we plan to investigate further in the future for the exact same reasons the reviewer stated here.

Overall, we understand the reviewer's comment and we will therefore include the reasoning earlier in the text so as to make the purpose more understandable to readers.

• Page 14 lines 240-253 and Figure 6: I would find it helpful if the abbreviations from Table 1 were used in this paragraph, which analyses Figure 6 (even though a sort of synthesis is presented in Table 2). I find it difficult to remember acronyms and abbreviations, so I had to go back and forth between the figure, the text and Table 1.

We agree with the reviewer. Initially, we tried to avoid repetition and including yet more information in this already dense paragraph. However, we will follow the reviewer's advice and add relevant abbreviations there.

• Table 2: How are potential and actual evapotranspiration obtained? Is it really important to include both in the table?

Both potential and actual evapotranspiration are S-HYPE model outputs. In our case, potential evapotranspiration is calculated based on mean temperature and a land use dependent rate parameter. An additional parameter adjusts the potential evaporation rate depending on the season. Regarding actual evapotranspiration, it is calculated by a linear function depending on soil moisture and it ranges between 0 and the potential evaporation value (when water content exceeds field capacity).

We included both in the table since, originally, we had the intention to include a short analysis based on the Budyko framework. However, since most catchments in Sweden are energy limited, it did not have much explanatory power.

We agree with the reviewer that it is not necessary here to present both parameters and we will therefore remove the potential evapotranspiration column in the revised manuscript.

• Page 16 line 268: Do you have any possible explanation why the cluster (5) with the highest general skill also have the largest spread? Is it possible that those two things (skill and spread) are related? What I mean is that if the skill is assessed by the CRPS and the CRPS is very sensitive to spread, then maybe the high skill is (at least in part) a consequence of this high spread? In any case, I think it would be interesting if the authors could provide a possible explanation.

Results from cluster 5 are indeed interesting. The forecasts in the cluster 5 catchments generally show the highest skill (for all lead times) among all cluster groups, yet results are widely spread. In this paper we conclude that the forecast skill is strongly linked to the various hydrological regimes (see also a more detailed investigation in Pechlivanidis et al. 2020), and hence we argue that the answer is within a deeper understanding of the hydrological signatures in cluster 5. As we state in P14 L241-242, the catchments in cluster 5 are characterized by a high baseflow contribution (BFI), a slow response to precipitation (Flash) and a generally small intra-annual variability (DPar). In Figure 6a we observe that although the mean values for RLD (rising limb density) are below the 33rd percentile of this signature (which represent 'below normal' signature values); however the boxplot

for RLD driven by all 4355 catchments in cluster 5 indicates high variability, with some catchments experience 'normal' RLD values and yet some others even higher than the 66th percentile of this signature. Consequently this indicates that some catchments in cluster 5 despite their high baseflow contribution experience sharp increases in their hydrographs, which is an indication of low skill as seen in Figure 5 (CRPSS and RLD are strongly, but negatively, correlated). We will explain the above argument for the large spread in cluster 5 in the revised manuscript.

• Page 18 lines 316-326: You mention the idea of using more sophisticated data assimilation techniques, such as the EnKF, but I think it would also be worth mentioning the possibility of assimilating other observations than streamflow, for instance soil moisture and/or snow water equivalent. This has been done in some studies (e.g. Huan et al. 2017), but there are still not that many in direct relation to seasonal forecasting.

In the revised manuscript we will acknowledge the possibility of assimilating other observations and refer to relevant studies such as Huan et al. 2017 or Musuuza et al. 2020, which is already cited in the manuscript.

• Page 19 lines 335-337: "This exercise shows that the regulation routines in . . ." There I finally found the justification for including regulated rivers in the study. I think this should be expressed earlier in the manuscript, around lines 105-120. At the moment, the explanations provided in lines 105-120 remain too general and it is hard to understand what it is that you want to test by including regulated rivers. At lines 335-337 it becomes clear, but it is too late.

Please see the previous comment on the same issue for a description on the reasoning behind this as well as the planned modifications to the revised manuscript.

References:

Day, G. (1985). Extended Streamflow Forecasting Using NWSRFS, J. Wat. Res. Plan. Mgmt., 10.1061/(ASCE)0733-9496(1985)111:2(157), 157-170

MeiBner et al. 2017 (already cited in the manuscript)

Baker S.A., Wood A.W. and Rajagopalan B. (2019). Developing Subseasonal to Seasonal Climate Forecast Products for Hydrology and Water Management, Journal of the American Water Resources Association, 55(4), 1024-1037.

Slater L.J., Villarini G., Bradley A.A. and Vecchi G.A. (2019) A dynamical statistical framework for seasonal forecasting in an agricultural watershed, Climate Dynamics, 53(12), 7429-7445.

Bazile R., Boucher M-A, Perreault L. And Leconte R. (2017) Verification of ECMWF System 4 for seasonal hydrological forecasting in a northern climate, Hydrology and Earth System Sciences, 21, 5747–5762.

Huang C., Newman A.J., Clark M.P., Wood A.W. and Zheng X. (2017) Evaluation of snow data assimilation using the ensemble Kalman filter for seasonal streamflow prediction in the western United States, Hydrology and Earth System Sciences, 21, 635–650.

Musuuza, J. L., Gustafsson, D., Pimentel, R., Crochemore, L. and Pechlivanidis, I. (2020): Impact of Satellite and In Situ Data Assimilation on Hydrological Predictions, Remote Sens., 12(5), 811, doi:10.3390/rs12050811.

Pechlivanidis, I. G., Crochemore, L., Rosberg, J. and Bosshard, T. (2020): What Are the Key Drivers Controlling the Quality of Seasonal Streamflow Forecasts?, Water Resour. Res., 56(6), doi:10.1029/2019WR026987.

---

## Author Comment (AC3) · 18 Dec 2020

**Authors' response to interactive comment by Reviewer #3 Shawn Harrigan**

Black text: Reviewer comment

Blue text: Authors' response

Girons Lopez et al. evaluate the forecast skill of theEnsemble Streamflow Prediction (ESP) method using the SMHI operational configuration of the S-HYPE hydrological model for providing seasonal streamflow forecasts across Sweden. They generate a set of ESP reforecasts at 25 ensemble members each for 39,493 catchments, initialised 4 times per month over the 36-year period from 1981-2016. The hydrological model, and reforecasts, are run at a daily time-step and are primarily aggregated to weekly averaged streamflow, out to a 6-month forecast horizon; a number of different temporal averages are also tested, from weekly to 24 weeks. For the 539 catchments with streamflow and water lever observations, an additional simple autoregressive algorithm was applied to correct raw modelled streamflow output prior to generation of the reforecasts. The probabilistic skill of ESP reforecasts was benchmarked using the Continuous Ranked Probability Skill Score (CRPSS) against a probabilistic (25 ensemble member) streamflow climatology (called "historical streamflow"in the paper) with modelled streamflow simulations as proxy observations (called "modelled reality" in the paper), or where available (i.e. 539 stations) in situ streamflow observations (called "observed reality" in the paper), as reference. Results show that ESP is skilful up to 3 months ahead for the most of Sweden. The strength of skill varies widely across the country in space and time and has shown to be linked to a number of key hydrological signatures (15 explored in total). Similar to previous work, ESP skill was highest in slowly responding baseflow-dominated (or high BFI) catchments and least skilful for flashy catchments. Seven unique clusters of similar hydrological behaviour were identified using k-means clustering and ESP skill summarised for catchments within each cluster.

I found this paper very interesting with a comprehensive ESP reforecast experimental design (i.e. long reforecast period, many forecast start dates, large sample of catchments, and cross validation used). It has the clear purpose of providing the scientific foundation for when and where the ESP forecast method is, and importantly is not, appropriate to use in operational seasonal forecasting across Sweden. The paper goes on to explore the potential sources of ESP skill based on correlation with hydrological signatures, while it's arguably a stretch to call correlation a formal attribution, the analysis nonetheless reveals interesting drivers and patterns of skill across the country, including the poor skill from catchments with high human disturbance (e.g. reservoirs). The paper is well structured and written with very good Figures. The analysis presented in Figure 7 is particularly insightful, and an innovative way of presenting forecast skill by clusters of similarly responding catchments. I offer below suggestions on areas where the paper could be expanded and highlight where clarifications are necessary, but these are all minor.

Therefore, I strongly recommend Girons Lopez et al. to be published in HESS. It adds to the growing literature benchmarking the skill of the ESP method with a clear application within operational seasonal forecasting at the national scale in Sweden.

We thank Dr Shawn Harrigan for his valuable comments and suggestions that will undoubtedly help us improve our manuscript. Below we reply to each of these and explain how we will incorporate them into the manuscript.

**Main comments**

1.) It would be useful to have these parts of the methods expanded/clarified:

a. Pg 3 L89-90: While there is a link to the general website to download the streamflow observations in the "Data availability" section, there is little detail for the reader on if all 539 stations are available in near-real time, which would be necessary to understand the transferability of forecast skill results to operational forecasts in future. Also, are all stations available for the full 1981 to 2016 period for calculation of KGE in Fig. 1 and for calculating the historical streamflow benchmark forecast? Were most of all these stations used for calibrating the configuration of S-HYPE used in the study?

The reviewer raises a valid point here, as hydrological observations are seldom available for long, overlapping periods across a large number of stations. The stations we used in this study are the ones being used operationally (and therefore collecting and sending data in real or near-real time) by the SMHI's service at the time of performing the analysis.

New stations are regularly added and some of the existing ones may be dismantled, but the sample is fairly similar to the one used for model calibration, as a recalibration effort for the most recent version of the model (the one we are using in this analysis) was performed recently.

Station availability and data quality is actually quite complex as, even if a large percentage of stations belong to SMHI, a significant part are external stations. Nevertheless, SMHI performs quality controls on all observations.

In short, because of the reasons listed above, data availability varies greatly among different stations. Nevertheless, most of them have over 20 years of data. Nonetheless, as the reviewer points out, the transferability of forecast skill results to operational forecasts in the future may need to be carefully assessed if there are significant changes in station availability.

In the revised version of the manuscript we will include a figure showing the periods with available observations for all stations (in the appendix) and we will include a sentence in Section 2.1 stating that data are assimilated from different stations at different times since data availability is not the same throughout the different stations for the 1981-2016 period.

b. Pg 5 L116-123: I find the AR correction interesting, but there is very little detail on how it was applied within the current experimental design, and perhaps even if it was implemented in such a way that is as consistent as feasibly possible to the configuration that is/will be implemented operationally?

The implementation of the AR correction in this reanalysis is indeed as close as possible to the operational setup. For instance, let us consider the case of a catchment which has observations throughout the analysis period. For each ESP initialisation, the model outputs are corrected up to the day before forecast initialisation. Then, as observations are theoretically no longer available, the model output correction starts from the latest correction value and exponentially decreases with time until the model outputs become clean simulation results (the rate of decrease is controlled by a model parameter). In the revised version of the manuscript, we will include this paradigmatic detailed description of the AR method.

c. Pg 5 Sect.2.3: The exact reforecast size is mixed between Sect. 2.2 and Sect. 2.3 and the reader has to try piece it together, it would be good if summarised. My understanding is that the reforecast dataset used has the following size: 39,493 catchments; 1728 start dates (4 start dates per month x 12 months x 36-year reforecast period (1981-2016)); weekly averaged streamflow out to 6 month forecast horizon at 25 ensemble members each?

That is correct. In the revised version of the manuscript we will rework the methods section to ensure that the description of how forecasts are generated is contained in a single section (Section 2.2). We will consequently also rename this section to "Hydrological modelling and forecasting".

d. Pg 5 L130-134: I think it could be confusing to refer to the probabilistic streamflow climatology benchmark forecast as "historical streamflow" because historical streamflow could more generally be interpreted by readers as the reference observations. I think it's more informative to be explicit about the type of benchmark forecast used for benchmarking skill (here, you indeed choose climatology which is the most appropriate given the seasonal forecast horizon).

We agree with the reviewer in that terminology should be used in a restrictive sense, as otherwise it could lead to misinterpretations. We will ensure that the appropriate term (streamflow climatology) is used throughout the text in the revised manuscript.

2.) Pg 7 L172-173: I'm not sure this is the correct conclusion from my interpretation of Fig. 2b and Fig. 3. It looks like skill initialised at the start of March (light green in Fig. 2b) is higher than any of the winter months, at least for the 1 week forecast horizon. This is confirmed in the map for 1 March in Fig. 3 for 1 week. Can you please clarify?

That is correct. The highest skill is indeed achieved for reforecasts initialised on 1 March (CRPSS above 0.8). The period with highest skill for forecast week 1 is actually between 8 December and 1 March, which we simplified in the text as "for forecasts initialised in winter" (Line 172). After 1 March the skill already decreases noticeably. This may be explained by the hydrological regimes of a large part of Swedish catchments, which generally start to increase in April-May, in addition to the general lack of precipitation in winter and early March. We will modify these lines to be more accurate.

3.) One of the key advantages of benchmarking ESP over Sweden is the opportunity to explore the role snow accumulation and melting has on controlling ESP skill. I can't help but think there's an additional piece of the puzzle missing in attributing ESP skill. While hydrological signatures are useful, e.g. baseflow index (BFI), there is not much discussion in the paper on the hydrological

processes within those catchments that are the source of ESP skill, based on information content and hence memory in the initial hydrological conditions. For example, a key question missing from the analysis is do catchments with a large contribution of streamflow from snow melt provide high skill when initialised around the snowmelt season? In practice, a catchment can have a high BFI due to several slowly responding processes (e.g. large groundwater/soil storage, snow, lakes, or a combination). I do not request this analysis is done, but it would be good to hear the authors' opinion and perhaps it could be worked into the discussion on the (initial hydrological condition) sources of ESP skill in Sweden.

We thank the reviewer for this interesting point. Unfortunately we did not calculate the contribution of streamflow from snow melt, and hence we cannot explicitly explore the role of snow accumulation/melting on ESP skill. Only in a case study investigation over the Umeälven river basin (snow dominated and heavily regulated system for hydropower production), we recently showed that assimilation of a snow water equivalent satellite-based product, particularly over the winter and spring seasons, significantly increased the streamflow forecasting skill. Note that this is still unpublished, whilst a manuscript is under preparation with an expected submission in early 2021.

Moreover, we agree with the reviewer that the definition of river memory can be a combination of processes, such as groundwater/baseflow contribution, snow accumulation/melting, and hydrograph dampening from lakes. Snow processes tend to define the river memory only seasonally (for example, precipitation in the form of snow in early December will be accumulated and further released as melting during the spring flood period), and hence the role of snow on ESP skill is expected to have a seasonal pattern too. This view will be mentioned in the last paragraph of Section 4.1 (Discussion).

**Technical comments**

4.) Pg 3 L69: Not sure "spread to other actors" is clear. A suggestion is: "ESP seasonal forecasts are produced operationally but have not been used widely in real-world applications due to lack of information on their skill...", or something to that effect?

We thank the reviewer for his suggestion. We will change this passage in the revised manuscript accordingly.

5.) Pg 4 L96-98: Can you please confirm the time scale the KGE was calculated, I presume it was calculated at daily time step from 1981-2016?

That is correct, the KGE values were calculated based on a forward run at a daily time step for the entire analysis period (1981-2016) using the S-HYPE model without station correction. We will clarify this in the revised version of the manuscript.

6.) Pg 4 Fig.1: Could you please add into the caption or text what exactly is shown in Fig. 1b and c in the coloured shapes, I presume it's the river network, or is it the river network downstream from an observed gauge only?

The coloured shapes in Fig. 1 correspond to those catchments that are being significantly corrected by observations over the entire analysis period (Fig. 1b) and that have a significant degree of

regulation (Fig. 1c). Even if they do not show the river network directly, they correspond quite well with it, since most stations and dams are located along watercourses. One can actually see the difference between both at the border with Finland (north-eastern part of the country): even if the Torne river there is not regulated (it is not shown in Fig. 1c), model outputs are still corrected - yet to a small degree - using the observations gathered from the stations along the river (which can be seen in Fig. 2b). We will clarify this in the text.

7.) Pg 20 L365: Suggest changing "reliable" to "skilful", as reliability was not explicitly evaluated.

We thank the reviewer for pointing this out. This is correct and we will therefore ensure that the appropriate term (i.e. skilful) is used throughout the revised version of the manuscript.

8.) Pg 21 L396-399: "sys", "bench" and "pft" more typically subscript, not superscript (i.e. $CRPS^{sys}$ should be $CRPS_{sys}$). Also, CRPSS values can range from 1 to $-\infty$, not "low negative values".

We will include the proposed modifications to the revised version of the manuscript.

---

## Author Response (AR2)

Dear HESS Editor Jim Freer,

Thank you again for your efforts with our manuscript. We have now addressed the points you raised and included most of the comments into the text. Below we provide a point-by-point reply to each of the comments as well as an explanation on how we included them into the text (blue text).

Additionally, we have identified and fixed other minor issues in the manuscript such as a missing affiliation for one of the co-authors, a broken cross-reference, or the missing "Acknowledgements" section.

With the aforementioned modifications we hope that this contribution meets the quality requirements to be published in Hydrology and Earth System Sciences.

Kind regards,

Marc Girons Lopez, Louise Crochemore and Ilias Pechlivanidis

**Authors' response to HESS Editor Jim Freer**

Dear Authors,

Thank you for your changes to the manuscript and responses to 3 comprehensive assessments by the reviewers, all who were positive to your paper. I have no problem with the fact that these minor corrections have been well tackled and you have developed considerable changes to the manuscript, your due diligence is appreciated.

We would like to thank the editor for the positive feedback.

One point though, that I was surprised did not come out in the reviews, but I would like personally to tackle you on. Namely that there seems a general disconnect in the paper that I think needs to be better discussed. The issue, as I see it, is as follows (and I know some of these matters you could note for a number of other hydrological modelling papers):

The first two points raised by the editor are very interesting. Yet, to our knowledge, there is no study that robustly addresses them. Hence, we are limited on only providing our individual views on the points below, supported by a set of references.

1.  KGE appears to be your benchmark for the basis of the forecast model. Clearly KGE has certain properties of evaluation to streamflow. You do not anywhere in the paper state why models based on maximizing KGE are valuable for your forecasting objectives - in fact this whole matter (objectives over experimental design) is hardly discussed anywhere. A model with different calibration statistics will clearly be different and thus be different as the underlying vehicle for the forecast analysis. Please can this be discussed in a meaningful way, so why these choices?

    This is a very good point. We start by saying that the S-HYPE model setup is the one used operationally by SMHI and not a version tuned to a specific model objective and/or set of user needs. It is also important to note that the model developments in the S-HYPE hydrological setup have been continuous since its first implementation in 2008. This operational setup has been used in early warning services, hydropower decision-making, water resources management etc. and hence its parameterization is not towards a (set of) characteristic(s) of the hydrograph (e.g. high flows for flood forecasting).

    Additionally, S-HYPE parameters have not been exclusively optimized using the KGE metric. The various model upgrades included model evaluations (calibration and verification periods) using, among other metrics, the NSE, KGE, relative bias, and multi-objective combinations between NSE and biases, whilst quite extensive effort was given into visual evaluation and manual fine-tuning of the model parameters. Here, we decided to select the KGE as a metric to communicate model performance, since after the Gupta et al. (2009) article, KGE has been considered as a benchmark in hydrological modelling. For S-HYPE the median KGE value for more than 530 stations is 0.79, which indicates a high model adequacy (to our knowledge, there is no other national hydrological setup with that high performance over that many stations). More specifically, this shows that different properties in the hydrograph (timing,

volume and variability) are well represented. The stations that show poor KGE values (<0.2; again, this threshold is subjective and driven by our own experience), are usually subject to dam/reservoir regulations, for which characteristics such as timing and variability are almost impossible to capture.

We addressed this comment by adding these considerations to Section 2.2 of the manuscript.

2. Then we move to the statistics on the output where you have table 2 showing a whole bunch of hydrological signatures used to explore a 15 dimensional space of forecast skill. Again there is a big disconnect here to any intelligent discussion of the objectives of the forecasts and why these signatures, and these signatures alone, have value. Indeed if at all why they should be considered (as it seems they are) equally weighted in the forecast assessments (do you even have an appreciation of their individual sensitivity to how good or bad the forecasts are?). Again I really think it should be explained to the reader why this table and why these metrics and do they really all have value to the types of the forecasts you want?

I am often frustrated at the 'many hydrological signatures' means a good jib has been one but without justification for their use. So we should justify this better, to explain the experimental designs we use. Again a different set of metrics or more or less of these signatures would generate potentially different results, and who is to say what best highlights 'what you need' for a given forecasting objective... And as I stated before why indeed a more comprehensive treatment of the model hindcast skill using various calibration metrics to test different periods and magnitudes of flow, were not used?

This is a very good point, which, to our knowledge, has not been addressed by previous investigations. As McMillan et al. (2017) stated, there is still a lack of consensus on a comprehensive set of hydrologic signatures to be used by the hydrological community.

Please further note that an investigation of the sensitivities to the selected hydrological signatures is not within the objectives of this article. Here, we aim to identify links between forecast skill and commonly used hydrological signatures. Our selection of signatures is driven by previous literature on hydrological classifications (Euser et al., 2013; Viglione et al., 2013), and by applications lead by scientists at SMHI oriented towards process understanding (Pechlivanidis and Arheimer 2015; Kuentz et al., 2017) and forecasting skill attribution (Pechlivanidis et al. 2020). We are also aware of other hydro-climatic clustering investigations (i.e. Knoben et al., 2018). Yet, to our knowledge, there is no investigation that guides the community towards a unique set of signatures for identifying hydrologic similarity (for example, Westerberg et al., (2016) only approached this topic from the view of uncertainties in hydrological signatures).

Furthermore, it is important to note here that, in our paper, we firstly assessed the link between forecast skill and individual hydrological signatures (step 1). To further generalize the insights from step 1, we investigated the link between forecast skill and hydrological regimes, as these are defined by the clustering of the signatures. We assume that the selected signatures are robust enough to guide the analysis towards the correct identification of hydrologically similar river systems.

Regarding the last question, please see our response to point 1.

We addressed this comment by adding these considerations to Sections 2.4 and 4.3 of the manuscript.

3. Finally I also want to clarify your use of a scale for KGE in Figure 1. Can you explain why you have set the scale to zero? KGE does not have the same properties as NS Efficiency where 0 is the mean predictor, in fact the mean predictor is -0.41 Knoben, W. J. M., J. E. Freer, and R. A. Woods (2019), Technical note: Inherent benchmark or not? Comparing Nash-Sutcliffe and Kling-Gupta efficiency scores, Hydrology and Earth System Sciences, 23(10), 4323-4331. for an explanation of this (please note I am not at all trying to get you to cite our paper). I just think you want to explain better why your figure has chosen certain limits.

Please note that in Figure 1a the lower limit is not 0. Yet, all stations with KGE values lower than 0 are represented in grey colour. We took the decision to group all stations (in total 8 stations out of 539) with KGE < 0, since the paper's focus is not on digging deeper on historical model performance and its spatial variability. As we explain in the discussion, poor model performance is observed in river systems that are regulated, since regulation schemes are almost impossible to fully represent. In such cases, model performance is thus poor in terms of timing (correlation coefficient) and variability (alpha term in KGE), which negatively affect the KGE values. Consequently, we do not believe that increasing the scale would add any value to the paper. On the contrary, it may distract the reader from the main objective of Fig. 1a, which is to show that KGE is very high for most stations.

**References**

Euser, T., Winsemius, H. C., Hrachowitz, M., Fenicia, F., Uhlenbrook, S., & Savenije, H. H. G. (2013). A framework to assess the realism of model structures using hydrological signatures. Hydrology and Earth System Sciences, 17(5), 1893–1912. https://doi.org/10.5194/hess-17-1893-2013

Kuentz, A., Arheimer, B., Hundecha, Y., & Wagener, T. (2017). Understanding hydrologic variability across Europe through catchment classification. Hydrology and Earth System Sciences, 21(6), 2863–2879. https://doi.org/10.5194/hess-21-2863-2017

Knoben, W. J. M., Woods, R. A., & Freer, J.E. (2018). A quantitative hydrological climate classification evaluated with independent streamflow data. Water Resources Research, 54, 5088–5109.https://doi.org/10.1029/2018WR022913

McMillan, H., Westerberg, I. and Branger, F.: Five guidelines for selecting hydrological signatures, Hydrological Processes, 31(26), 4757–4761, https://doi.org/10.1002/hyp.11300, 2017.

Pechlivanidis, I. G., & Arheimer, B. (2015). Large-scale hydrological modelling by using modified PUB recommendations: the India-HYPE case. Hydrology and Earth System Sciences, 19, 4559–4579. https://doi.org/10.5194/hess-19-4559-2015

Pechlivanidis, I. G., Crochemore, L., Rosberg, J., & Bosshard, T. (2020). What are the key drivers controlling the quality of seasonal streamflow forecasts? Water Resources Research, 56, e2019WR026987. https://doi.org/10.1029/2019wr026987

Viglione, A., Parajka, J., Rogger, M., Salinas, J. L., Laaha, G., Sivapalan, M., & Blöschl, G. (2013). Comparative assessment of predictions in ungauged basins – Part 3: Runoff signatures in Austria. Hydrology and Earth System Sciences, 17(6), 2263–2279. https://doi.org/10.5194/hess-17-2263-2013

Westerberg, I. K., Wagener, T., Coxon, G., McMillan, H. K., Castellarin, A., Montanari, A., & Freer, J. (2016). Uncertainty in hydrological signatures for gauged and ungauged catchments. Water Resources Research, 52, 1–19. https://doi.org/10.1002/2015WR017635